# On the Robustness of Mechanism Design under Total Variation Distance

**Anuran Makur**
Purdue University
amakur@purdue.edu

**Marios Mertzanidis**
Purdue University
mmertzan@purdue.edu

**Alexandros Psomas**
Purdue University
apsomas@cs.purdue.edu

**Athina Terzoglou**
Purdue University
aterzogl@purdue.edu

## Abstract

We study the problem of designing mechanisms when agents' valuation functions are drawn from unknown and correlated prior distributions. In particular, we are given a prior distribution $\mathcal{D}$, and we are interested in designing a (truthful) mechanism that has good performance for all "true distributions" that are close to $\mathcal{D}$ in Total Variation (TV) distance. We show that DSIC and BIC mechanisms in this setting are strongly robust with respect to TV distance, for any bounded objective function $\mathcal{O}$, extending a recent result of Brustle et al. ([BCD20], EC 2020). At the heart of our result is a fundamental duality property of total variation distance. As direct applications of our result, we (i) demonstrate how to find approximately revenue-optimal and approximately BIC mechanisms for weakly dependent prior distributions; (ii) show how to find correlation-robust mechanisms when only "noisy" versions of marginals are accessible, extending recent results of Bei et. al. ([BGLT19], SODA 2019); (iii) prove that prophet-inequality type guarantees are preserved for correlated priors, recovering a variant of a result of Dütting and Kesselheim ([DK19], EC 2019); (iv) give a new necessary condition for a correlated distribution to witness an infinite separation in revenue between simple and optimal mechanisms, complementing recent results of Psomas et al. ([PSCW22], NeurIPS 2022); (v) give a new condition for simple mechanisms to approximate revenue-optimal mechanisms for the case of a single agent whose type is drawn from a correlated distribution that can be captured by a Markov Random Field, complementing recent results of Cai and Oikonomou ([CO21], EC 2021).

## 1 Introduction

Mechanism design studies optimization in strategic settings. The designer's task is to create a mechanism that interacts with strategic agents, each with their own, private preferences over the mechanism's output. The inability to provide meaningful guarantees for important objectives — such as revenue — when studying mechanism design problems through the lens of worst-case analysis, has motivated the study of Bayesian mechanisms. In the Bayesian setting, there is a probability distribution, typically known to the designer, from which agents' private information — their *types* — is drawn, and the designer seeks to maximize an objective function in expectation over the randomness of the types, and, at the same time, incentivize the agents to report their type truthfully.

While we have greatly deepened our understanding of mechanism design under the Bayesian setting, when taking this approach to practice, it is natural to ask what happens if the designer has only partial information about the agents' type distributions. In recent years, a growing literature studies

the *robustness* of mechanisms with respect to inaccurate priors. The term "robustness" can mean many things in this context, such as mechanism design using only sampling access from the underlying prior [CR14, HMR15, DHP16, MR16, CD17, GHZ19, GW21], or mechanism design using only parametric knowledge of the underlying prior [AMDW13, Car17, GL18, BGLT19, GPTD23]. Arguably, the most ambitious line of work in this thread strives to, given a prior distribution $\mathcal{D}$, design a mechanism that provides good guarantees simultaneously for all "true distributions" $\widehat{\mathcal{D}}$ that are close to $\mathcal{D}$ under some notion of statistical distance. Recent, compelling positive results [BS11, CD17, DK19, LLY19, BCD20] show that this endeavor is, in fact, possible.

In this work, we demonstrate that a large class of such robustness results can be obtained from a fundamental duality property of total variation distance. We distill this key property into various forms that are directly applicable to mechanism design, and then illustrate how they imply a variety of robustness results for both Dominant Strategy Incentive Compatible (DSIC) and Bayesian Incentive Compatible (BIC) mechanisms. In particular, we recover a litany of known results, such as the revenue robustness of BIC mechanisms of [BCD20] and robustness of prophet inequalities of [DK19], as well as prove several new results, such as a new necessary condition for a prior distribution to exhibit a large gap between simple and optimal mechanisms (complementing recent results of [PSCW22]).

**Our Contributions.**  We consider the problem of a designer that wants to design a mechanism that maximizes a bounded objective function $\mathcal{O}$, when allocating $m$ items to $n$ strategic agents. Agent $i$ has a private type $t_i \in \mathcal{T}_i$, drawn from a probability distribution $\mathcal{D}_i$, which specifies her value for every subset of items $S \subseteq [m]$. We write $\mathcal{D}$ for the joint distribution over agents' types.

We start in Section 3 by proving a simple, but crucial lemma, Lemma 2, about non-truthful mechanisms in this setting. Lemma 2 states that the expected performance, with respect to the objective $\mathcal{O}$, of a (possibly non-truthful) mechanism $\mathcal{M}$ is stable with respect to small changes in *total variation (TV) distance* to the underlying prior distribution. This lemma follows from the argument for Kantorovich-Rubinstein duality of TV distance [LPW09, Vil09].

Appropriate applications of Lemma 2 allow us to prove our first robustness result: *DSIC mechanisms are strongly robust*. Specifically, given a mechanism $\mathcal{M}$ that is ex-post individually rational (ex-post IR), DSIC, and an $\alpha$ approximation to the optimal mechanism for a distribution $\mathcal{D}$ (w.r.t. the objective $\mathcal{O}$), we construct an ex-post IR and DSIC mechanism $\widehat{\mathcal{M}}$ that is agnostic to $\widehat{\mathcal{D}}$ and $\alpha$ approximates, under distribution $\widehat{\mathcal{D}}$, the performance of the *optimal* mechanism for $\widehat{\mathcal{D}}$ (w.r.t. $\mathcal{O}$) minus a small error which depends on the TV distance, $d_{\mathsf{TV}}\left(\mathcal{D}, \widehat{\mathcal{D}}\right)$, of $\mathcal{D}$ and $\widehat{\mathcal{D}}$ (**Theorem 1**). Notably, $\mathcal{D}$ and $\widehat{\mathcal{D}}$ need not be product distributions, and if they have the same support, $\widehat{\mathcal{M}}$ is the same mechanism as $\mathcal{M}$.

We proceed to study the robustness of BIC mechanisms. As opposed to the DSIC case, the BIC property is not maintained under small perturbations of a prior, or even by small changes in the support of a distribution. Rather surprisingly, however, in Lemma 4 we show that a mechanism $\mathcal{M}$ that is BIC w.r.t. a distribution $\mathcal{D}$, is also approximately BIC w.r.t. a distribution $\widehat{\mathcal{D}}$, where the approximation depends on the distance between $\mathcal{D}$ and $\widehat{\mathcal{D}}$. Combined with Lemma 2, we show our main robustness result for BIC mechanisms (**Theorem 2**): *a mechanism $\mathcal{M}$ that is BIC w.r.t. $\mathcal{D}$ is approximately BIC w.r.t. $\widehat{\mathcal{D}}$, and its performance for an objective function $\mathcal{O}$ is similar under $\mathcal{D}$ and $\widehat{\mathcal{D}}$, as long as $\mathcal{D}$ and $\widehat{\mathcal{D}}$ have the same support*. Compared to the corresponding robustness result of [BCD20], Theorem 2 only holds if $\mathcal{D}$ and $\widehat{\mathcal{D}}$ have the same support. Furthermore, $\mathcal{M}$ is not $\varepsilon$-BIC (as in [BCD20]), but $(\varepsilon, q)$-BIC (see Section 2). On the other hand, our robustness results hold for *arbitrary objectives*, not just revenue, and *arbitrary distributions*, not just product distributions.

In Section 4 we show numerous applications of our robustness framework. Specifically, we give five applications. First, in Section 4.1, we extend our strong DSIC robustness to BIC mechanisms, albeit only for the revenue objective and product distributions. That is, we show that given a distribution $\mathcal{D}$ that is close to a product distribution $\mathcal{D}^p$, approximately optimal (w.r.t. revenue) and BIC mechanisms for $\mathcal{D}^p$ can be transformed into approximately optimal and approximately-BIC mechanisms for $\mathcal{D}$ (**Theorem 3**). En route, we prove a self-reduction, that constructs an $\varepsilon'$-BIC mechanism w.r.t. $\mathcal{D}$, given an $(\varepsilon, q)$-BIC mechanism w.r.t. $\mathcal{D}$, that might be of independent interest.

In Section 4.2 we study the correlation-robust framework of [BGLT19] (first introduced in [Car17]). At a high level, we are given the marginals of $n$ dependent agents for a single item. The goal is to design a mechanism that maximizes (among all feasible mechanisms) the minimum, over all possible

joint distributions consistent with the given marginals, expected revenue. We extend our robustness results to this setting (**Theorem 4**), and get implications for the max-min revenue performance of sequential posted prices and the Lookahead auction when the given marginals are inaccurate.

In Section 4.3 we consider prophet inequalities. In the simplest version of this problem, we are shown $n$ non-negative numbers, $t_1, \ldots, t_n$, one at a time, and upon arrival we need to decide (immediately and irrevocably) whether we should accept $t_i$, or keep going. $t_i$s are drawn independently from known distributions $\mathcal{D}_i$. A celebrated result, known as the prophet inequality, states that a simple threshold policy $\tau$ ("pick the first $x_i$ bigger than $\tau$") has expected reward at least half of a prophet, who knows the values of all $x_i$s in advance (and can therefore get reward $\max_i t_i$). Prophet inequalities have numerous applications in mechanism design, and are the main tool in the analysis of posted price mechanisms. Observing that prophet inequalities correspond to sequential posted prices, which are DSIC mechanisms, and applying Lemma 2 for the welfare objective, we get robustness of prophet inequalities (**Corollary 1**); as a special case of our application we recover a variant of a result of Dütting and Kesselheim [DK19] on the TV robustness of prophet inequalities.

Finally, in Section 4.4 we apply our robustness framework to study revenue gaps between simple and optimal mechanisms. Here, [BCKW15, HN19] construct correlated distributions whose revenue is infinite, but such that simple mechanisms (e.g., those with finite menus) cannot extract a lot of revenue. Recently, Psomas et al. [PSCW22] provide (arguably, complex) conditions that a distribution $\mathcal{D}$ should satisfy in order to witness such large gaps; our framework readily provides new, simple, and necessary conditions (**Corollary 2**), complementing the results of [PSCW22]. On a similar note, Cai and Oikonomou [CO21] escape the negative results of [BCKW15, HN19] by considering distributions described by a Markov Random Field (MRF). Their revenue guarantees for simple mechanisms are controlled by $\Delta$, a parameter of the MRF, that is determined by how much the value of an item can be influenced by values of other items. We show that a bound on the TV distance to a product distribution also suffices to provide guarantees on the revenue of simple mechanisms (**Proposition 2**). Furthermore, under conditions on the MRF, bounds on $\Delta$ imply bounds on this distance (**Proposition 4**), but the other direction is not necessarily true (**Proposition 3**); thus, getting bounds on the TV distance of an MRF to a product distribution is a meaningful endeavor.

**Related Work.** A number of recent works study robustness in mechanism design; see [Car17] for a survey. The paper (thematically and technically) closest to ours is [BCD20], which proves the robustness of the revenue objective for BIC mechanisms under various notions of statistical distance, including TV and Prokhorov, as well as Lévy and Kolmogorov for single parameter settings. Specifically, given a distribution $\mathcal{D}$, [BCD20] show how to construct a mechanism that is (approximately) BIC, and performs well, when executed on any distribution $\widehat{\mathcal{D}}$ that is close to $\mathcal{D}$. Here, we focus on TV distance, and recover their result for this case, slightly extending it for the case of revenue, and significantly extending it under some minor assumptions. Dütting and Kesselheim [DK19] study prophet inequalities with inaccurate priors. Specifically, given a product distribution $\times_{i \in [n]} \mathcal{D}_i$, [DK19] study policies that perform well when executed on a product distribution $\times_{i \in [n]} \widehat{\mathcal{D}}_i$, such that $\mathcal{D}_i$ is close to $\widehat{\mathcal{D}}_i$ for all $i \in [n]$, under various statistical distances. As an application of our robustness for DSIC mechanisms, we recover and extend to non-product distributions their result for TV distance, for the special case of sequential posted prices.

A related, but different, approach is to assume sample access to the underlying distribution [CR14, HMR15, DHP16, MR16, CD17, GHZ19, GW21]. In this line of work, the goal is bound the number of samples necessary to design a near-optimal mechanism, or, given a fixed number of samples, design the best mechanism possible. Robustness results of the former style, i.e., the current paper or [BCD20], sometimes imply sample complexity results, e.g., by arguing that using the samples, one can learn a distribution that is close to the real distribution, and then applying robustness results [BCD20]. We conjecture that similar results can be shown using our robustness framework, combined with estimation in TV distance [HJW15, DG85]. Another line of work [AMDW13, Car17, GL18, BGLT19, GPTD23] assumes partial knowledge of the true prior distribution, e.g., its mean or the marginals of a correlated distribution. The goal is to find the mechanism that (approximately) maximizes the worst-case performance with respect to the missing details (e.g., the CDF consistent with the mean or the joint distribution that respects the given marginals). Our results have implications for such settings as well. Finally, certain works consider the robustness of pathological examples in mechanism design, e.g., constructions of distributions that have infinite revenue gaps between simple and optimal mechanisms [BCKW15, HN19]. For example, [PSW19] uses the lens of smoothed

analysis ([ST04]) to reason about the robustness of the [HN19] constructions. Our framework has implications about these constructions; specifically, we give a new necessary condition for a distribution to be "pathological," complementing a recent result of [PSCW22].

From a probability theory perspective, we heavily use coupling techniques. Couplings are a general proof technique in probability theory with several historically notable uses [Doe38, Sko56, Kan60, Str65, Dud68, Wic70, Dob70] (also see [Vil09, Kal21]). Given two marginal distributions, the basic idea of coupling is to construct a consistent joint distribution on a common probability space in order to deduce certain relationships between the marginals. The key coupling used in this work — Dobrushin's optimal coupling — minimizes the probability that two random variables with given marginal distributions are different, and has been historically utilized to develop sharper results on Markov chain ergodicity, cf. [Dob70, Dob71, Gri75]. It turns out that this optimal coupling also defines total variation distance, cf. [LPW09, Vil09]. Such distances exhibit Kantorovich-Rubinstein duality and are characterized by the maximal difference of expected values with respect to the given marginal distributions [Kan60, LPW09, Vil09]. In this work, we distill how optimal couplings and duality for total variation can be used in yet another setting: mechanism design.

## 2 Preliminaries

We examine the problem of a central designer who seeks to create a mechanism that maximizes an objective function given some prior knowledge of the universe. Consider a set $[n] \triangleq \{1, \ldots, n\}$ of selfish agents and a finite set $[m] \triangleq \{1, \ldots, m\}$ of items. Each agent $i \in [n]$ has a type $t_i$ belonging to a set $\mathcal{T}_i$ of possible types. We assume that, for all $i \in [n]$, there exists a special type $\perp \in \mathcal{T}_i$, interpreted as the option of not participating in the designer's mechanism. Let $\mathcal{T} = \mathcal{T}_1 \times \cdots \times \mathcal{T}_n$. We use $\mathcal{X}$ to denote the set of all possible allocations of the items. (In particular, the sets $\mathcal{X}$ and $\mathcal{T}_i$ are typically standard Borel spaces, e.g., finite-dimensional Euclidean spaces. Hence, our analysis with couplings and total variation distance in the sequel do not require measure theoretic arguments.)

The goal of the designer is to construct a mechanism $\mathcal{M} = (x, p)$ which consists of (1) an allocation rule $x : \mathcal{T} \mapsto \Delta(\mathcal{X})$, which maps reported types $(t_1, \ldots, t_n) \in \mathcal{T}$ to a distribution over allocations, and (2) a payment function $p : \mathcal{T} \mapsto [-H, H]^n$ which maps reported types to (bounded) payments for each agent, for some fixed constant $H > 0$. We say that a mechanism $\mathcal{M} = (x, p)$ is *defined* on types $\mathcal{T}$ if the domain of $x$ and $p$ is $\mathcal{T}$. We write $\mathcal{M}(t) = (x(t), p(t))$ for the outcome, i.e., the allocation and payments, of mechanism $\mathcal{M}$ on input $t \in \mathcal{T}$.

Each agent $i$ has a valuation function $v_i : \mathcal{T}_i \times \mathcal{X} \mapsto [0, H]$, which specifies their value for an allocation, given their type. We assume that agents are quasi-linear, i.e., the *utility* of agent $i$, with type $t_i$, for an allocation $A$ and a payment $p_i$ is equal to $v_i(t_i, A) - p_i$. We overload notation and write $t_i(\mathcal{M}(t_i, t_{-i}))$ for the utility of agent $i$ with type $t_i \in \mathcal{T}_i$ for the outcome of the mechanism $\mathcal{M}$ on input $(t_i, t_{-i})$. We use $\mathcal{F}$ for the set of possible outcomes of a mechanism $\mathcal{M}$. We assume that there exists a probability distribution $\mathcal{D}$ supported on $\mathcal{T}$ from which agents' types are drawn.

**Mechanism Design Considerations.** When faced with a mechanism $\mathcal{M}$, each agent $i$ reports a type $b_i \in \mathcal{T}_i$ to the mechanism. We aim to design mechanisms that (approximately) incentivize agents to report their true types, so we typically have that $b_i = t_i$, for all $i \in [n]$. Given a mechanism $\mathcal{M}$, we write $u_i^{\mathcal{M}}(t_i \leftarrow t_i', t_{-i})$ for the difference in utility of agent $i$ when she reports $t_i' \in \mathcal{T}_i$ instead of her true type $t_i \in \mathcal{T}_i$, and all other agents report according to $t_{-i} \in \mathcal{T}_{-i}$. That is, $u_i^{\mathcal{M}}(t_i \leftarrow t_i', t_{-i}) = t_i(\mathcal{M}(t_i, t_{-i})) - t_i(\mathcal{M}(t_i', t_{-i}))$.

We consider four, increasingly weaker, notions of incentive compatibility. First, we say that a mechanism $\mathcal{M}$ is *Dominant Strategy Incentive Compatible* (henceforth, DSIC) if an agent is better off reporting her true type, no matter what other agents report, i.e., for all $i \in [n]$, every type $t_i \in \mathcal{T}_i$, possible misreport $t_i' \in \mathcal{T}_i$, and types $t_{-i} \in \mathcal{T}_{-i}$ for the remaining agents, it holds that $u_i^{\mathcal{M}}(t_i \leftarrow t_i', t_{-i}) \geq 0$. Second, a mechanism $\mathcal{M}$ is *Bayesian Incentive Compatible* (henceforth BIC) with respect to a distribution $\mathcal{D}$, if an agent is better off reporting her true type in expectation over the other agents' reports, i.e. if for all $i \in [n]$, every type $t_i \in \mathcal{T}_i$ and possible misreport $t_i' \in \mathcal{T}_i$, it holds that $\mathbb{E}_{t_{-i} \sim \mathcal{D}_{-i}|t_i} [u_i^{\mathcal{M}}(t_i \leftarrow t_i', t_{-i})] \geq 0$. Third, a mechanism $\mathcal{M}$ is *ε-BIC* w.r.t. $\mathcal{D}$ if for all $i \in [n]$, and all $t_i, t_i' \in \mathcal{T}_i$, it holds that $\mathbb{E}_{t_{-i} \sim \mathcal{D}_{-i}|t_i} [u_i^{\mathcal{M}}(t_i \leftarrow t_i', t_{-i})] \geq -\varepsilon$. Fourth, a mechanism $\mathcal{M}$ is *(ε, q)-BIC* w.r.t. $\mathcal{D}$ if it is ε-BIC with probability at least $1 - q$, i.e., if for all $i \in [n]$, and all

$t_i, t_i' \in \mathcal{T}_i$, $\Pr_{t_i \sim \mathcal{D}_i} \left[ \mathbb{E}_{t_{-i} \sim \mathcal{D}_{-i}|t_i} \left[ u_i^{\mathcal{M}}(t_i \leftarrow t_i', t_{-i}) \right] \geq -\varepsilon \right] \geq 1 - q$. An $(\varepsilon, 0)$-BIC mechanism is simply $\varepsilon$-BIC, and a 0-BIC is simply BIC.

Finally, a mechanism $\mathcal{M}$ is *ex-post Individually Rational* (henceforth ex-post IR) if for every agent $i \in [n]$, every type $t_i \in \mathcal{T}_i$ and types $t_{-i} \in \mathcal{T}_{-i}$ for the remaining agents, it holds that $t_i (\mathcal{M}(t_i, t_{-i})) \geq 0$.

**The Designer's Objective.** The designer has an objective function $\mathcal{O}(.,.) : \mathcal{T} \times \mathcal{F} \mapsto [a, b]$ that takes as input agents' (reported) types $t \in \mathcal{T}$ and mechanism outcomes $\mathcal{M}(t) \in \mathcal{F}$ (noting that the mechanism's outcome might be a randomized allocation) and outputs a real number in the interval $[a, b]$. Let $V = b - a$. The task of the designer is to find an ex-post IR and truthful mechanism (under one of the aforementioned notions of truthfulness) that maximizes this objective function in expectation over the randomness of $\mathcal{D}$. We denote the optimal value of the objective $\mathcal{O}$ under distribution $\mathcal{D}$ by $OPT_{\mathcal{O}}(\mathcal{D})$. We specify in context whether this is with respect to DSIC, or BIC, or $\varepsilon$-BIC mechanisms. A mechanism $\mathcal{M}_{\mathcal{D}}^\alpha$ is an $\alpha$ approximation to the optimal mechanism under $\mathcal{D}$, with respect to $\mathcal{O}$, if $\mathbb{E}_{t \sim \mathcal{D}} \left[ \mathcal{O}(t, \mathcal{M}_{\mathcal{D}}^\alpha(t)) \right] \geq a \, OPT_{\mathcal{O}}(\mathcal{D})$.

Some of our results hold for arbitrary, bounded objectives $\mathcal{O}$. Two objectives of specific interest to us will be welfare and revenue. The welfare objective, denoted by $Val(.)$, is simply the sum of agents' valuations of an outcome, i.e. $Val(t, \mathcal{M}(t)) = \sum_{i \in [n]} v_i(t_i, x_i(t))$, where $x_i(t)$ is the allocation of agent $i$ in the outcome $\mathcal{M}(t)$. The revenue objective, denoted by $Rev(.)$, is the sum of agents' payments, i.e. $Rev(t, \mathcal{M}(t)) = \sum_{i \in [n]} p_i(t)$, where $p_i(t)$ is the payment of agent $i$ in the outcome $\mathcal{M}(t)$. We often overload notation and write $Rev(\mathcal{M}, \mathcal{D})$ for the expected revenue of mechanism $\mathcal{M}$ under distribution $\mathcal{D}$, i.e. $Rev(\mathcal{M}, \mathcal{D}) = \mathbb{E}_{t \sim \mathcal{D}} \left[ Rev(t, \mathcal{M}(t)) \right]$. To maintain consistency with the (vast) literature on mechanism design we further denote the optimal revenue as $Rev(\mathcal{D})$, and the optimal welfare as $Val(\mathcal{D})$, under distribution $\mathcal{D}$.

**Statistical Distance.** Throughout the paper, we are interested in how mechanisms behave under different distributions that are not "too far" from each other. Our notion of distance in this paper is total variation distance.

**Definition 1** (Total Variation Distance). *The* total variation (TV) distance *between any two probability distributions $P$ and $Q$ on a sample space $\Omega$ is defined as*

$$d_{\mathsf{TV}}(P, Q) \triangleq \sup_{E \subseteq \Omega} |P(E) - Q(E)|,$$

*where the supremum is over all Borel measurable subsets $E \subseteq \Omega$, and $P(E)$ (resp. $Q(E)$) denotes the probability of the event $E$ with respect to the distribution $P$ (resp. $Q$).*

We note that $\Omega$ is either discrete or a measurable subset of a finite-dimensional Euclidean space in our analysis, and hence, it is always a standard Borel space. For any probability distributions $P$ and $Q$ on $\Omega$, let $\Pi(P, Q)$ be the (non-empty) set of all *couplings* of $P$ and $Q$, i.e., all joint probability distributions $\gamma = P_{X,Y}$ of two random variables $X, Y \in \Omega$ such that the marginal distributions are $P_X = P$ and $P_Y = Q$, respectively. The following definition will be of utility in the sequel [Dob70, LPW09].

**Definition 2** (Optimal Coupling). *For any two probability distributions $P$ and $Q$ on $\Omega$, we define the* optimal coupling *of $P$ and $Q$ as the joint distribution $\gamma^* = \arg\min_{\gamma \in \Pi(P,Q)} \mathbb{E}_{(X,Y) \sim \gamma} \left[ \mathbb{1}\{X \neq Y\} \right]$ of two random variables $(X, Y)$ that has marginal distributions $P$ and $Q$ of $X$ and $Y$, respectively, and minimizes the probability that $X$ is different to $Y$.*

We note that such an optimal coupling always exists [Dob70, Vil09]. The next lemma presents several useful characterizations of TV distance including an optimal coupling representation (which demonstrates how TV distance is a Wasserstein distance with respect to the discrete metric [Vil09]); see [Mak19, Section 2.2.1] and [PW22, Theorem 7.7] for a compilation of other characterizations.

**Lemma 1** (Equivalent Characterizations of TV Distance [Str65, Dob70, LPW09, Vil09]). *For any two probability distributions $P$ and $Q$ on $\Omega$, we have*

$$\begin{aligned} d_{\mathsf{TV}}(P, Q) &= \frac{1}{2} \|P - Q\|_1 \\ &= \min_{\gamma \in \Pi(P,Q)} \Pr_{(X,Y) \sim \gamma} [X \neq Y] \\ &= \max_{f:\Omega \to [-\frac{1}{2}, \frac{1}{2}]} \mathbb{E}_{X \sim P} [f(X)] - \mathbb{E}_{Y \sim Q} [f(Y)], \end{aligned}$$

where $\|P - Q\|_1$ is the $\mathcal{L}^1$-*distance between* $P$ *and* $Q$, *the second equality is the optimal coupling characterization which minimizes* $\Pr_{(X,Y)\sim\gamma}[X \neq Y] = \mathbb{E}_{(X,Y)\sim\gamma}[\mathbb{1}\{X \neq Y\}]$ *over all couplings* $\gamma$ *of* $P$ *and* $Q$, *and the third equality is the* Kantorovich-Rubinstein dual *characterization which takes the maximum over all (measurable) functions* $f : \Omega \to \mathbb{R}$ *bounded by* $\sup_{t\in\Omega} |f(t)| \leq \frac{1}{2}$.

# 3 Robustness of Mechanisms Under Total Variation Distance

In this section, we prove our robustness results for DSIC and BIC mechanisms. Missing proofs throughout the section are deferred to Appendix A.

At the heart of our approach lies the following lemma, which shows that, even for correlated prior distributions, assuming truthful bidding, a mechanism's performance with respect to an arbitrary objective function $\mathcal{O}$ is stable under small perturbations to the prior.

**Lemma 2.** *Let* $P$ *and* $Q$ *be two arbitrary probability distributions supported on* $\mathcal{T}$ *and let* $\mathcal{M}$ *be any mechanism. Assuming truthful bidding, for all objective functions* $\mathcal{O}(.,.) \in [a,b]$, *letting* $V = b - a$, *it holds that* $\mathbb{E}_{t\sim P}[\mathcal{O}(t, \mathcal{M}(t))] - \mathbb{E}_{t'\sim Q}[\mathcal{O}(t', \mathcal{M}(t'))] \leq V\, d_{\mathsf{TV}}(P, Q)$.

We note that $\mathcal{M}$ in the statement of Lemma 2 may not be DSIC or BIC, for neither $P$ nor $Q$.

*Proof of Lemma 2.* The objective function is lower bounded by $a$ and upper bounded by $b$; thus, we have that for any $t, t' \in \mathcal{T}$, $\mathcal{O}(t, \mathcal{M}(t)) - \mathcal{O}(t', \mathcal{M}(t')) \leq V\, \mathbb{1}\{t \neq t'\}$. Since this inequality holds for all $t, t' \in \mathcal{T}$, it also holds after taking an expectation with respect to any coupling $\gamma$ of $P$ and $Q$, and specifically for the optimal coupling $\gamma^*$ between $P$ and $Q$ (see Definition 2):

$$\mathbb{E}_{(t,t')\sim\gamma^*}[\mathcal{O}(t, \mathcal{M}(t)) - \mathcal{O}(t', \mathcal{M}(t'))] \leq V\, \mathbb{E}_{(t,t')\sim\gamma^*}[\mathbb{1}\{t \neq t'\}]. \tag{1}$$

Using the second form of Lemma 1, the RHS of (1) is equal to $V\, d_{\mathsf{TV}}(P, Q)$. For the LHS of (1), using linearity of expectation, the fact that $\mathcal{O}(t, \mathcal{M}(t))$ does not depend on $t'$, and the fact that $\mathcal{O}(t', \mathcal{M}(t'))$ does not depend on $t$, we have that $\mathbb{E}_{(t,t')\sim\gamma^*}[\mathcal{O}(t, \mathcal{M}(t)) - \mathcal{O}(t', \mathcal{M}(t'))] = \mathbb{E}_{t\sim P}[\mathcal{O}(t, \mathcal{M}(t))] - \mathbb{E}_{t'\sim Q}[\mathcal{O}(t', \mathcal{M}(t'))]$. Putting everything together, we get the desired inequality. $\square$

In the remainder of this section, we show that this lemma can be used to prove strong robustness results for DSIC and BIC mechanisms.

## 3.1 DSIC Mechanisms

Our main robustness result for DSIC mechanisms is stated as follows.

**Theorem 1** (Robustness for DSIC). *Let* $\mathcal{D}$ *and* $\widehat{\mathcal{D}}$ *be two arbitrary distributions supported on* $\mathcal{T}$, *respectively, such that* $d_{\mathsf{TV}}\left(\mathcal{D}, \widehat{\mathcal{D}}\right) \leq \delta$, *and let* $\mathcal{O}(.,.) \in [a,b]$ *be an objective function. Let* $\mathcal{M}_{\mathcal{D}}^\alpha$ *be an ex-post IR, DSIC, and* $\alpha$-*approximate mechanism (under* $\mathcal{D}$), *with respect to the optimal (under* $\mathcal{D}$) *ex-post IR and DSIC mechanism for* $\mathcal{O}$. *Then, letting* $V = b - a$, *it holds that*

$$\mathbb{E}_{t\sim\widehat{\mathcal{D}}}[\mathcal{O}(t, \mathcal{M}_{\mathcal{D}}^\alpha(t))] \geq \alpha\, OPT_{\mathcal{O}}(\widehat{\mathcal{D}}) - (1 + \alpha)V\delta,$$

*where* $OPT_{\mathcal{O}}(\widehat{\mathcal{D}})$ *is the performance of the optimal (for* $\mathcal{O}$) *DSIC mechanism.*

*Proof.* We apply Lemma 2 twice. First, choosing $P = \mathcal{D}$, $Q = \widehat{\mathcal{D}}$, and $\mathcal{M} = \mathcal{M}_{\mathcal{D}}^\alpha$, Lemma 2 implies

$$\mathbb{E}_{t\sim\mathcal{D}}[\mathcal{O}(t, \mathcal{M}_{\mathcal{D}}^\alpha(t))] - \mathbb{E}_{t'\sim\widehat{\mathcal{D}}}[\mathcal{O}(t', \mathcal{M}_{\mathcal{D}}^\alpha(t'))] \leq V\, d_{\mathsf{TV}}\left(\mathcal{D}, \widehat{\mathcal{D}}\right) \leq V\delta.$$

Now, let $\mathcal{M}_{\widehat{\mathcal{D}}}^*$ be the optimal (under $\widehat{\mathcal{D}}$) ex-post IR and DSIC mechanism for $\mathcal{O}$. Using the definition of $\mathcal{M}_{\mathcal{D}}^\alpha$, and re-arranging we have

$$\begin{aligned}
\mathbb{E}_{t'\sim\widehat{\mathcal{D}}}[\mathcal{O}(t', \mathcal{M}_{\mathcal{D}}^\alpha(t'))] &\geq \mathbb{E}_{t\sim\mathcal{D}}[\mathcal{O}(t, \mathcal{M}_{\mathcal{D}}^\alpha(t))] - V\delta \\
&\geq \alpha\, \mathbb{E}_{t\sim\mathcal{D}}[\mathcal{O}(t, \mathcal{M}_{\mathcal{D}}^*(t))] - V\delta \\
&\geq \alpha\, \mathbb{E}_{t\sim\mathcal{D}}\left[\mathcal{O}(t, \mathcal{M}_{\widehat{\mathcal{D}}}^*(t))\right] - V\delta,
\end{aligned}$$

where the last inequality is because $\mathcal{M}_{\widehat{\mathcal{D}}}^*$ is feasible (i.e. ex-post IR and DSIC) for $\mathcal{D}$. Second, choosing $P = \widehat{\mathcal{D}}$, $Q = \mathcal{D}$, and $\mathcal{M} = \mathcal{M}_{\widehat{\mathcal{D}}}^*$, by Lemma 2, plus re-arranging, we have that $\mathbb{E}_{t' \sim \mathcal{D}} \left[ \mathcal{O}(t', \mathcal{M}_{\widehat{\mathcal{D}}}^*(t')) \right] \geq \mathbb{E}_{t \sim \widehat{\mathcal{D}}} \left[ \mathcal{O}(t, \mathcal{M}_{\widehat{\mathcal{D}}}^*(t)) \right] - V \delta$ Combining these inequalities we have:

$$\mathbb{E}_{t \sim \widehat{\mathcal{D}}} [\mathcal{O}(t, \mathcal{M}_D^\alpha(t))] \geq \alpha \, \mathbb{E}_{t \sim \widehat{\mathcal{D}}} \left[ \mathcal{O}(t, \mathcal{M}_{\widehat{\mathcal{D}}}^*(t)) \right] - (1 + \alpha) V \delta. \qquad \square$$

Intuitively, Theorem 1 states that, if a mechanism is approximately optimal for $\mathcal{D}$, then it is also approximately optimal for all $\widehat{\mathcal{D}}$ that are close in total variation distance, paying a small additive error. Note that neither $\mathcal{D}$ nor $\widehat{\mathcal{D}}$ needs to be a product distribution. That is, if $\mathcal{D}$ and $\widehat{\mathcal{D}}$ have the same support, a DSIC mechanism $\mathcal{M}_{\mathcal{D}}^\alpha$ performs approximately-optimally under $\widehat{\mathcal{D}}$. Finally, notice that in the above theorem, we assume that the $\mathcal{D}$ and $\widehat{\mathcal{D}}$ share the same support; in Appendix A.1 we relax this assumption, and show how to modify the mechanism $\mathcal{M}_{\mathcal{D}}^\alpha$, in a way that is agnostic to $\widehat{\mathcal{D}}$, and provide exactly the same guarantee.

### 3.2 Bayesian Incentive Robustness

In this section, we study BIC mechanisms. As opposed to DSIC mechanisms, arguing about whether a BIC mechanism remains BIC after perturbing the prior distribution is a lot more involved. Our goal in this section is to prove that the BIC property degrades gracefully as a function of the TV distance, even for arbitrary objectives, albeit, with two small technical caveats (compared to the DSIC robustness): (1) $\mathcal{D}$ and $\widehat{\mathcal{D}}$ must share the same support, and (2) the incentive guarantees of our mechanisms also degrade. The second requirement is necessary; in Section 4.1 we show how to bypass the first requirement for the revenue objective, recovering a slightly stronger version of a TV robustness result of Brustle et al. [BCD20].

First, we need the following Markov-like technical lemma that when two joint distributions $P_{X,Y}$, $Q_{X,Y}$ are close in TV distance, then, with high probability, the conditional distributions $P_{Y|X=x}$, $Q_{Y|X=x}$ are also close in TV distance.

**Lemma 3.** *Let $P_{X,Y}$, $Q_{X,Y}$ be two probability distributions for the (possibly multivariate) random variables $X$ and $Y$. Let $P_{Y|X=x}$ (resp. $Q_{Y|X=x}$) be the probability distribution of $P_{X,Y}$ (resp. $Q_{X,Y}$) conditioned on $X = x$, and let $Q_X$ be the marginal probability distribution of $X$ as dictated by $Q_{X,Y}$. Then, for all $q \in [0,1]$, $\Pr_{x \sim Q_X} \left[ d_{\mathsf{TV}} \left( P_{Y|X=x}, Q_{Y|X=x} \right) > \frac{2 \, d_{\mathsf{TV}}(P_{X,Y}, Q_{X,Y})}{q} \right] \leq q.$*

Next, we prove that a BIC mechanism for $\mathcal{D}$ is $(\varepsilon, q)$-BIC for $\widehat{\mathcal{D}}$, assuming that $\mathcal{D}$ and $\widehat{\mathcal{D}}$ have the same support and small TV distance.

**Lemma 4.** *Let $\mathcal{D}$ and $\widehat{\mathcal{D}}$ be two probability distributions supported on $\mathcal{T}$, with $d_{\mathsf{TV}} \left( \mathcal{D}, \widehat{\mathcal{D}} \right) \leq \delta$. If $\mathcal{M}$ is an ex-post IR and BIC mechanism w.r.t. $\mathcal{D}$ then it is also an ex-post IR and $(\frac{8H\delta}{q}, q)$-BIC mechanism w.r.t. $\widehat{\mathcal{D}}$, for all $q \in [0,1]$.*

To prove the above lemma we leverage Lemma 3. We know that with high probability the perception of each agent over the distributions of the rest of the agents is very close under $\mathcal{D}$ and $\widehat{\mathcal{D}}$. That is why, if agent $i$ cannot gain by misreporting under $\mathcal{D}$, then with high probability, she cannot significantly increase her utility by misreporting under $\widehat{\mathcal{D}}$.

As a direct implication of Lemma 2 and Lemma 4 we get our main robustness result for BIC.

**Theorem 2** (Robustness for BIC). *Let $\mathcal{D}$ and $\widehat{\mathcal{D}}$ be two arbitrary distributions supported on $\mathcal{T}$, such that $d_{\mathsf{TV}} \left( \mathcal{D}, \widehat{\mathcal{D}} \right) \leq \delta$, and let $\mathcal{O}(.,.) \in [a,b]$ be an objective function. Let $\mathcal{M}_{\mathcal{D}}$ be a mechanism that is ex-post IR and BIC w.r.t. $\mathcal{D}$. Then $\mathcal{M}_{\mathcal{D}}$ is also ex-post IR and $\left( \frac{8H\delta}{q}, q \right)$-BIC w.r.t. $\widehat{\mathcal{D}}$, for all $q \in [0,1]$. Also, letting $V = b - a$, it holds that $\mathbb{E}_{t \sim \widehat{\mathcal{D}}} [\mathcal{O}(t, \mathcal{M}_{\mathcal{D}}(t))] \geq \mathbb{E}_{t \sim \mathcal{D}} [\mathcal{O}(t, \mathcal{M}_{\mathcal{D}}(t))] - V\delta.$*

Comparing to the corresponding result of Brustle et al. [BCD20], Theorem 2 only holds if $\mathcal{D}$ and $\widehat{\mathcal{D}}$ have the same support. Also, the guarantee on incentives is weaker: $\mathcal{M}_{\mathcal{D}}$ is not $\varepsilon$-BIC (as in [BCD20]), but $(\varepsilon, q)$-BIC. However, our robustness results holds for *arbitrary objectives*, not just revenue, and *arbitrary distributions*, not just product distributions.

**On Tightness of Robustness Results.** Regarding Lemma 2, it is known that $\mathbb{E}_{X \sim P}[f(X)] - \mathbb{E}_{X \sim Q}[f(X)] \leq d_{\mathsf{TV}}(P, Q)$ for all functions $f$ bounded by $1/2$ (see Lemma 1), and equality holds for the function $f^*(x) = 1/2$ if $P(x) \geq Q(x)$ and $f^*(x) = -1/2$ otherwise. We can use this to show that equality holds for Lemma 2 when the objective function and the mechanism, combined, look like this function, i.e., $f^*(x) = \mathcal{O}(x, \mathcal{M}(x))$ (with appropriate re-scaling when $V$ is not 1). This yields a sufficient condition for tightness of Lemma 2.

Regarding Theorem 1, it is straightforward to construct a tight example for the case of revenue and welfare. For instance, for the case of a single agent, letting $\mathcal{T} = [0, V]$, consider the case that the distribution $P$ is a point mass at $V$, and distribution $Q$ takes the value $V$ with probability $1 - \delta$, and zero otherwise. The TV distance between $P$ and $Q$ is $\delta$. Consider the simple mechanism $\mathcal{M}$ that posts a price of $V$. Its revenue/welfare under $P$ is $V$, and its revenue/welfare under $Q$ is $(1 - \delta)V$. The main issue with generalizing to arbitrary objectives is that a worst-case arbitrary objective can do something uninteresting, e.g., take the value $c$ no matter what, where naturally our result is not tight.

Regarding Theorem 2, tightness of the revenue objective follows from the aforementioned tightness of Theorem 1 for revenue. Tightness for the BIC guarantee follows from the following example. Consider the single item case where two identical bidders have valuation 1 with probability 0.5 and valuation 2 with probability 0.5 independently from each other. Now consider the mechanism where the first bidder always takes the item if he bids 2 and pays 1.5. When he bids 1 and the second bidder bids 1, he again takes the item and and now pays 1. Finally, if the first bidder bids 1 and the second bidder bids 2, then the second bidder takes the item and pays 2. For the specific distribution we selected, it is easy to check that this mechanism is BIC. Now assume that the second bidder's distribution changes to having valuation 1 with probability $0.5 + \varepsilon$ and 2 with probability $0.5 - \varepsilon$. The TV distance between the two distributions is exactly $\varepsilon$. However our mechanism is no longer BIC. Whenever the first bidder's valuation is 2, if he reports truthfully, he will always get 0.5 utility. If he instead reports 1, he will make $0.5 + \varepsilon$ utility (on expectation). Taking into account this observation, we can see that our mechanism is now $\varepsilon$-BIC.

# 4 Applications

In this section, we show a number of applications of Lemma 2, and Theorems 1 and 2.

## 4.1 BIC Mechanisms

We start by showing applications of our robustness results for the revenue objective of BIC mechanisms. Our goal is to extend our DSIC robustness result (Theorem 1) to BIC mechanisms. That is, we'd like, given an approximately optimal BIC mechanism for a distribution $\mathcal{D}$, to get an approximately optimal and approximately BIC mechanism for distribution $\widehat{\mathcal{D}}$. We will achieve this goal for the revenue objective and product distributions; missing proofs can be found in Appendix B.

Towards our main result for this section, we prove the following lemma, which might be of independent interest. Intuitively, the lemma shows that one can turn an $(\varepsilon, q)$-BIC mechanism into a $O(\varepsilon + nqH)$-BIC mechanism, paying a small loss in revenue.

**Lemma 5.** *For any product distribution $\mathcal{D} = \times_{i \in [n]} \mathcal{D}_i$, given a mechanism $\mathcal{M}$ that is ex-post IR and $(\varepsilon, q)$-BIC w.r.t. $\mathcal{D}$, we can design a mechanism $\widehat{\mathcal{M}}$ that is ex-post IR and $O(\varepsilon + nqH)$-BIC w.r.t. $\mathcal{D}$, such that $Rev(\widehat{\mathcal{M}}, \mathcal{D}) \geq Rev(\mathcal{M}, \mathcal{D}) - nqV$.*

Our main theorem for this stage is stated as follows.

**Theorem 3.** *Let $\mathcal{D}$ be a probability distribution supported on $\mathcal{T}$, and let $\mathcal{D}^p$ be a product distribution such that $d_{\mathsf{TV}}(\mathcal{D}, \mathcal{D}^p) \leq \delta$. Let $\mathcal{M}_{\mathcal{D}^p}^{\alpha}$ be an ex-post IR, BIC, and $\alpha$-approximate mechanism (under $\mathcal{D}^p$), with respect to the revenue optimal (under $\mathcal{D}^p$) ex-post IR and BIC mechanism. Then, $\mathcal{M}_{\mathcal{D}^p}^{\alpha}$ is ex-post IR and $(\frac{8H\delta}{q}, q)$-BIC with respect to $\mathcal{D}$, for all $q \in (0, 1]$. Furthermore, $Rev(\mathcal{M}_{\mathcal{D}^p}^{\alpha}, \mathcal{D}) \geq \alpha \, OPT(\mathcal{D}) - O\left((1 + \alpha)V\sqrt{n\sqrt{\delta}}\right)$.*

In order to get the full benefits of Theorem 3, one needs the product distribution $\mathcal{D}^p$ that is the closest (in TV distance) to the original distribution $\mathcal{D}$. Let $\varepsilon$ be this optimal distance. The following proposition shows that, for every distribution $\mathcal{D}$, the distance of $\mathcal{D}$ to the product of its marginals is bounded by $(n + 1)\varepsilon$, and therefore, Theorem 3 holds for every distribution $\mathcal{D}$ for $\delta = (n + 1)\varepsilon$.

**Proposition 1.** *Let $\mathcal{D} = \times_{i \in [n]} \mathcal{D}_i$ be a product distribution supported on $\times_{i \in [n]} \mathcal{T}_i$, and let $\widehat{\mathcal{D}}$ be a joint distribution supported on $\times_{i \in [n]} \mathcal{T}_i$, whose marginal over $\mathcal{T}_i$ is exactly $\mathcal{D}_i$. Then, if there exists a product distribution $\mathcal{D}^p$ such that $d_{\mathsf{TV}}\left(\widehat{\mathcal{D}}, \mathcal{D}^p\right) \leq \varepsilon$, it holds that $d_{\mathsf{TV}}\left(\mathcal{D}, \widehat{\mathcal{D}}\right) \leq (n+1)\varepsilon$.*

## 4.2 Marginal Robustness

As a second application, we consider the setting of Bei et al. [BGLT19]. In this problem, we want to sell one item to $n$ dependent agents, and we only know the marginal distribution of each agent $i$. Overloading notation, let $t_i \in \mathcal{T}_i \subseteq \mathbb{R}_+$ be the valuation of agent $i$ for the item. There is a distribution $\mathcal{D}$, supported on $\times_{i \in [n]} \mathcal{T}_i$, from which agents' valuations are sampled from. For the sake of simplicity, we only argue about discrete distributions; however, our results easily extend to the continuous case. Let $\mathcal{D}_i$ be the marginal distribution of agent $i$, i.e., $\Pr_{t_i \sim \mathcal{D}_i}[t_i = v_i] = \sum_{v_{-i} \in \mathcal{T}_{-i}} \Pr_{t \sim \mathcal{D}}[t = (v_i, v_{-i})]$. The designer knows marginal distributions $\mathcal{D}_i$ for each $i$ but not $\mathcal{D}$. Given a set of marginal distributions $(\mathcal{D}_1, \cdots, \mathcal{D}_n)$ let $\Pi(\mathcal{D}_1, \cdots, \mathcal{D}_n)$ be the set of all distributions consistent with such marginals, i.e. $\Pi(\mathcal{D}_1, \cdots, \mathcal{D}_n) = \{\mathcal{D}' | \Pr_{t_i \sim \mathcal{D}_i}[t_i = v_i] = \sum_{v_{-i} \in \mathcal{T}_{-i}} \Pr_{t \sim \mathcal{D}'}[t = (v_i, v_{-i})], \forall i \in [n], \forall t_i \in \mathcal{T}_i\}$. Our goal is to find a mechanism $\mathcal{M}$ such that $\mathcal{M} = \arg\max_{\mathcal{M}'} \min_{\mathcal{D}' \in \Pi(\mathcal{D}_1, \cdots, \mathcal{D}_n)} \mathbb{E}_{t \sim \mathcal{D}'}[\mathcal{O}(t, \mathcal{M}'(t))]$ where $\mathcal{M}'$ is taken over all possible ex-post IR and DSIC mechanisms. We can prove the following theorem.

**Theorem 4.** *Given a set of marginals $(\mathcal{D}_1, \cdots, \mathcal{D}_n)$ and a DSIC and ex-post IR mechanism $\mathcal{M}^\alpha$ such that $\min_{\mathcal{D} \in \Pi(\mathcal{D}_1, \cdots, \mathcal{D}_n)} \mathbb{E}_{t \sim \mathcal{D}}[\mathcal{O}(t, \mathcal{M}^\alpha(t))] \geq \alpha \max_{\mathcal{M}'} \min_{\mathcal{D} \in \Pi(\mathcal{D}_1, \cdots, \mathcal{D}_n)} \mathbb{E}_{t \sim \mathcal{D}}[\mathcal{O}(t, \mathcal{M}'(t))]$, then for any set of marginals $(\mathcal{D}'_1, \cdots, \mathcal{D}'_n)$ such that for all $i \in [n]$, $d_{\mathsf{TV}}(\mathcal{D}_i, \mathcal{D}'_i) \leq \varepsilon$, it holds that $\min_{\mathcal{D}' \in \Pi(\mathcal{D}'_1, \cdots, \mathcal{D}'_n)} \mathbb{E}_{t \sim \mathcal{D}'}[\mathcal{O}(t, \mathcal{M}^\alpha(t))] \geq \alpha \max_{\mathcal{M}'} \min_{\mathcal{D}' \in \Pi(\mathcal{D}'_1, \cdots, \mathcal{D}'_n)} \mathbb{E}_{t \sim \mathcal{D}'}[\mathcal{O}(t, \mathcal{M}'(t))] - (1 + \alpha)n\varepsilon V$.*

The input to our problem is a set of marginals; however, we do not sample from these marginals to compute our objective. Instead, these marginals are used in order to derive a new distribution from which we will sample. Therefore, we cannot immediately "black box" the results we have shown until now. To prove Theorem 4 we first relate, and bound the "distance" between $\Pi(\mathcal{D}_1, \cdots, \mathcal{D}_n)$ and $\Pi(\mathcal{D}'_1, \cdots, \mathcal{D}'_n)$. Then, we need to re-prove arguments equivalent to the ones used for our robustness results so far, while taking into consideration the $\min\max$ nature of the problem. We postpone the formal proof of Theorem 4 to Appendix C.

[BGLT19] show that Sequential Posted Prices Mechanisms are a $4.78$-approximation and there exists a Lookahead Auction that is a $2$-approximation with respect to revenue for the above problem. Using Theorem 4, we can readily get implications for the robustness of those two auctions.

## 4.3 Prophet Inequalities

Here, we study the prophet inequality problem. Recall that in this problem, agents arrive over time; in the $i$-th step, we need to (immediately and irrevocably) decide on the allocation of agent $i$, whose type $t_i$ is drawn from a marginal distribution $\mathcal{D}_i$. Our goal is to design a prophet inequality: a policy that competes with the optimal in hindsight welfare maximizing allocation. It is known that, in fairly general domains, one can achieve this goal using posted prices, i.e., set a price $p_j$ for each item $j$, and let users pick their utility-maximizing subset of items [SC84, FGL14, RS17, KW19, ANSS19, DKL20, DFKL20]; see [Luc17] for a survey. Using Lemma 2 for the welfare objective we prove robustness for prophet inequalities, noting that posted price mechanisms are ex-post IR and DSIC.

**Corollary 1.** *For a posted price mechanism $\mathcal{M}$, and distributions $\mathcal{D}$, $\widehat{\mathcal{D}}$ such that $d_{\mathsf{TV}}\left(\mathcal{D}, \widehat{\mathcal{D}}\right) \leq \delta$, it holds that $Val(\mathcal{M}, \mathcal{D}) \geq Val(\mathcal{M}, \widehat{\mathcal{D}}) - V\delta$.*

Using Corollary 1 for $\mathcal{D} = \times_{i \in [n]} \mathcal{D}_i$ and $\widehat{\mathcal{D}} = \times_{i \in [n]} \widehat{\mathcal{D}}_i$ that are product distributions, we can get the TV robustness result of Dütting and Kesselheim [DK19] for sequential posted prices as a special case, noting that (1) if $d_{\mathsf{TV}}\left(\mathcal{D}_i, \widehat{\mathcal{D}}_i\right) \leq \varepsilon$, then $d_{\mathsf{TV}}\left(\mathcal{D}, \widehat{\mathcal{D}}\right) \leq n\varepsilon$, and (2) if valuations are normalized to $[0, 1]$ (as in [DK19]), $V = n$.

## 4.4 Gaps between Simple and Optimal Mechanisms

Here, we show applications of our robustness results to the study of simple and approximately optimal mechanisms. Missing proofs are deferred to Appendix D.

Motivated by numerous negative results for revenue optimal auctions (see [Das15] for a survey), a major research thread in mechanism design studies the performance of simple mechanisms [CHK07, CHMS10, CMS15, Yao15, RW15, CM16, CDW16, CZ17, KW19, BILW20]. Two canonical mechanisms that are considered simple in this literature are: (1) the mechanism that sells each item separately (and optimally), and (2) the mechanism that optimally sells all items as a grand bundle; let $SRev(\mathcal{D})$ and $BRev(\mathcal{D})$ be the expected revenue of these mechanisms, respectively, under prior $\mathcal{D}$. On the flip side, product distributions are known to witness simple mechanisms that are approximately optimal. For instance, Babaioff et. al. [BILW20] prove that for a single agent and a product distribution $\mathcal{D}^p$, $\max\{SRev(\mathcal{D}^p), BRev(\mathcal{D}^p)\} \geq \frac{1}{6}Rev(\mathcal{D}^p)$, even though the individual approximation factors for $SRev(\mathcal{D}^p)$ and $BRev(\mathcal{D}^p)$ are $O(1/\log(m))$ and $1/m$, respectively [HN13, LY13]. Note that in the single-agent context, by product distribution, we mean with respect to items. The following observation is an immediate implication of Theorem 1.

**Observation 1.** *Let $\mathcal{SM}$ be a family of mechanisms, such that $\mathbb{E}_{t \sim \widehat{\mathcal{D}}^p} \left[ \mathcal{O}(t, \mathcal{SM}_{\widehat{\mathcal{D}}^p}(t)) \right] \geq \alpha\, OPT_{\mathcal{O}}(\widehat{\mathcal{D}}^p)$ for some objective $\mathcal{O}$, where $\mathcal{SM}_{\widehat{\mathcal{D}}^p}$ is a mechanism in $\mathcal{SM}$ parameterized by the product distribution $\widehat{\mathcal{D}}^p$. Then, for any (possibly non-product) distribution $\mathcal{D}$ that is close to some product distributions $\mathcal{D}^p$, and specifically, $d_{\mathsf{TV}}(\mathcal{D}, \mathcal{D}^p) \leq \frac{\alpha OPT_{\mathcal{O}}(\mathcal{D})}{2(1+\alpha)V}$, we have that $\mathbb{E}_{t \sim \mathcal{D}}\left[\mathcal{O}(t, \mathcal{SM}_{\mathcal{D}^p}(t))\right] \geq \frac{\alpha}{2}OPT_{\mathcal{O}}(\mathcal{D})$.*

A construction of Hart and Nisan [HN13] shows that there exists a non-product distribution $\mathcal{D}$ such that $SRev(\mathcal{D}) \leq 2m$, $BRev(\mathcal{D}) \leq 2m$ but $Rev(\mathcal{D}) \to \infty$. Recently, [PSCW22] show necessary and sufficient conditions for a distribution $\mathcal{D}$ to exhibit such infinite gaps between the revenue of simple and optimal mechanisms. Unfortunately, these conditions are rather complex (namely, they are conditions on the existence of infinite sequences of points with certain properties). Here, leveraging Observation 1 for the revenue objective, we complement results of [PSCW22] by giving a new, simple necessary condition that a "pathological" construction must satisfy:

**Corollary 2.** *For a single agent and any distribution $\mathcal{D}$ such that $\frac{Rev(\mathcal{D})}{BRev(\mathcal{D})} \geq 2m$ it must be the case that for any product distribution $\mathcal{D}^p$, $d_{\mathsf{TV}}(\mathcal{D}, \mathcal{D}^p) \geq \frac{Rev(\mathcal{D})}{4V}$. Furthermore, if $\frac{Rev(\mathcal{D})}{SRev(\mathcal{D})} \in \Omega(\log(m))$ it must be the case that for any product distribution $\mathcal{D}^p$, $d_{\mathsf{TV}}(\mathcal{D}, \mathcal{D}^p) \geq \frac{Rev(\mathcal{D})}{4V}$.*

As a final application of our framework, we give new positive results on simple mechanisms.

**Proposition 2.** *Let $\mathcal{D}$ be a distribution supported on $\mathcal{T}$, and let $\mathcal{D}^p$ be a product distribution such that $d_{\mathsf{TV}}(\mathcal{D}, \mathcal{D}^p) < \delta$. Then, we have $\max\{SRev(\mathcal{D}), Brev(\mathcal{D})\} \geq \frac{1}{6}\, Rev(\mathcal{D}) - \frac{7}{6}H\delta$.*

Proposition 2 implies that the better of bundling and selling each item separately for a distribution $\mathcal{D}$ is a good approximation to $Rev(\mathcal{D})$, as long as $\mathcal{D}$ is close to a product distribution. Cai and Oikonomou [CO21] prove that the same mechanism is a good approximation for dependent distributions that can be captured by a Markov Random Field (MRF); see Appendix D for basic definitions regarding MRFs. The approximation ratio of [CO21] is controlled by $\Delta$, a parameter of the MRF that is determined by how much the value of an item can be influenced by the values of the other items. Specifically, [CO21] prove that $\max\{SRev(\mathcal{D}), BRev(\mathcal{D})\} \geq \frac{1}{12e^{4\Delta}}Rev(\mathcal{D})$. In Proposition 3, we prove that there exist distributions such that the object of interest for Proposition 2, i.e., the distance to a product distribution, is arbitrarily small, while the objective of interest for [CO21], the parameter $\Delta$, is arbitrarily large. At the same time, in Proposition 4 we show that when the MRF parameter $\Delta$ is bounded, we can bound the distance of a distribution $\mathcal{D}$ to a product distribution, for distributions that can be represented by MRFs that only have pairwise edges.

**Proposition 3.** *For any $0 < k < 1/2$, there exists a distribution $\mathcal{D}$ produced by an MRF with parameter $\Delta$ and a product distribution $\mathcal{D}^p$ such that $d_{\mathsf{TV}}(\mathcal{D}, \mathcal{D}^p) \leq 2k^2$ and $\Delta \geq \frac{1}{4}\log\left(\frac{1}{k}\right)$.*

**Proposition 4.** *Let $\mathcal{D}$ be a distribution produced by an MRF with only pairwise edges. Then, there exists a product distribution $\mathcal{D}^p$ such that $d_{\mathsf{TV}}(\mathcal{D}, \mathcal{D}^p) \leq \min\{\sqrt{m\Delta/4}, \sqrt{1 - e^{-m\Delta/2}}\}$, where $m$ is the number of items.*

## Acknowledgments

Alexandros Psomas is supported in part by an NSF CAREER award CCF-2144208, a Google Research Scholar Award, and a Google AI for Social Good award. Athina Terzoglou is supported in part by an NSF CAREER award CCF-2144208. Marios Mertzanidis is supported in part by a DOE award SC0022085, and NSF awards CCF-1814041, CCF-2209509, and DMS-2152687.

