# References

[AMDW13] Pablo Azar, Silvio Micali, Constantinos Daskalakis, and S Matthew Weinberg. Optimal and efficient parametric auctions. In *Proceedings of the twenty-fourth annual ACM-SIAM symposium on Discrete algorithms*, pages 596–604. SIAM, 2013.

[ANSS19] Nima Anari, Rad Niazadeh, Amin Saberi, and Ali Shameli. Nearly optimal pricing algorithms for production constrained and laminar bayesian selection. In *Proceedings of the 2019 ACM Conference on Economics and Computation*, pages 91–92, 2019.

[BCD20] Johannes Brustle, Yang Cai, and Constantinos Daskalakis. Multi-item mechanisms without item-independence: Learnability via robustness. In *Proceedings of the 21st ACM Conference on Economics and Computation*, EC '20, page 715–761, New York, NY, USA, 2020. Association for Computing Machinery.

[BCKW15] Patrick Briest, Shuchi Chawla, Robert Kleinberg, and S Matthew Weinberg. Pricing lotteries. *Journal of Economic Theory*, 156:144–174, 2015.

[BGLT19] Xiaohui Bei, Nick Gravin, Pinyan Lu, and Zhihao Gavin Tang. Correlation-robust analysis of single item auction. In *Proceedings of the Thirtieth Annual ACM-SIAM Symposium on Discrete Algorithms*, pages 193–208. SIAM, 2019.

[BH78] J. L. Bretagnolle and Catherine Huber. Estimation des densités: risque minimax. *Zeitschrift für Wahrscheinlichkeitstheorie und Verwandte Gebiete*, 47:119–137, 1978.

[BILW20] Moshe Babaioff, Nicole Immorlica, Brendan Lucier, and S Matthew Weinberg. A simple and approximately optimal mechanism for an additive buyer. *Journal of the ACM (JACM)*, 67(4):1–40, 2020.

[BS11] Dirk Bergemann and Karl Schlag. Robust monopoly pricing. *Journal of Economic Theory*, 146(6):2527–2543, 2011.

[Car17] Gabriel Carroll. Robustness and separation in multidimensional screening. *Econometrica*, 85(2):453–488, 2017.

[CD17] Yang Cai and Constantinos Daskalakis. Learning multi-item auctions with (or without) samples. In *2017 IEEE 58th Annual Symposium on Foundations of Computer Science (FOCS)*, pages 516–527. IEEE, 2017.

[CDW16] Yang Cai, Nikhil R. Devanur, and S. Matthew Weinberg. A duality based unified approach to bayesian mechanism design. In *Proceedings of the Forty-Eighth Annual ACM Symposium on Theory of Computing*, STOC '16, page 926–939, New York, NY, USA, 2016. Association for Computing Machinery.

[CHK07] Shuchi Chawla, Jason D. Hartline, and Robert Kleinberg. Algorithmic pricing via virtual valuations. In *Proceedings of the 8th ACM Conference on Electronic Commerce*, EC '07, pages 243–251, New York, NY, USA, 2007. ACM.

[CHMS10] Shuchi Chawla, Jason D Hartline, David L Malec, and Balasubramanian Sivan. Multi-parameter mechanism design and sequential posted pricing. In *Proceedings of the forty-second ACM symposium on Theory of computing*, pages 311–320. ACM, 2010.

[CM16] Shuchi Chawla and J Benjamin Miller. Mechanism design for subadditive agents via an ex ante relaxation. In *Proceedings of the 2016 ACM Conference on Economics and Computation*, pages 579–596. ACM, 2016.

[CMS15] Shuchi Chawla, David Malec, and Balasubramanian Sivan. The power of randomness in bayesian optimal mechanism design. *Games and Economic Behavior*, 91:297–317, 2015.

[CO21] Yang Cai and Argyris Oikonomou. On simple mechanisms for dependent items. In *Proceedings of the 22nd ACM Conference on Economics and Computation*, pages 242–262, 2021.

[COVZ21]  Yang Cai, Argyris Oikonomou, Grigoris Velegkas, and Mingfei Zhao. An efficient $\varepsilon$-bic to bic transformation and its application to black-box reduction in revenue maximization. In *Proceedings of the 2021 ACM-SIAM Symposium on Discrete Algorithms (SODA)*, pages 1337–1356. SIAM, 2021.

[CR14]  Richard Cole and Tim Roughgarden. The sample complexity of revenue maximization. In *Proceedings of the forty-sixth annual ACM symposium on Theory of computing*, pages 243–252, 2014.

[CZ17]  Yang Cai and Mingfei Zhao. Simple mechanisms for subadditive buyers via duality. In *Proceedings of the 49th Annual ACM SIGACT Symposium on Theory of Computing*, pages 170–183. ACM, 2017.

[Das15]  Constantinos Daskalakis. Multi-item auctions defying intuition? *ACM SIGecom Exchanges*, 14(1):41–75, 2015.

[DFK11]  Shahar Dobzinski, Hu Fu, and Robert D. Kleinberg. Optimal auctions with correlated bidders are easy. In *Proceedings of the Forty-Third Annual ACM Symposium on Theory of Computing*, STOC '11, page 129–138, New York, NY, USA, 2011. Association for Computing Machinery.

[DFKL20]  Paul Dutting, Michal Feldman, Thomas Kesselheim, and Brendan Lucier. Prophet inequalities made easy: Stochastic optimization by pricing nonstochastic inputs. *SIAM Journal on Computing*, 49(3):540–582, 2020.

[DG85]  Luc Devroye and László Györfi. *Nonparametric Density Estimation: The $L_1$ View*. Wiley Series in Probability and Mathematical Statistics. John Wiley & Sons, Inc., New York, NY, USA, 1985.

[DHP16]  Nikhil R Devanur, Zhiyi Huang, and Christos-Alexandros Psomas. The sample complexity of auctions with side information. In *Proceedings of the forty-eighth annual ACM symposium on Theory of Computing*, pages 426–439, 2016.

[DK19]  Paul Dütting and Thomas Kesselheim. Posted pricing and prophet inequalities with inaccurate priors. In *Proceedings of the 2019 ACM Conference on Economics and Computation*, EC '19, page 111–129, New York, NY, USA, 2019. Association for Computing Machinery.

[DKL20]  Paul Dütting, Thomas Kesselheim, and Brendan Lucier. An o (log log m) prophet inequality for subadditive combinatorial auctions. *ACM SIGecom Exchanges*, 18(2):32–37, 2020.

[Dob70]  Roland L. Dobrushin. Prescribing a system of random variables by conditional distributions. *Theory of Probability and Its Applications*, 15(3):469–497, 1970.

[Dob71]  Roland L. Dobrushin. Markov processes with a large number of locally interacting components: Existence of a limit process and its ergodicity. *Problemy Peredachi Informatsii*, 7(2):70–87, 1971.

[Doe38]  Wolfgang Doeblin. Exposé de la théorie des chaînes simples constantes de Markov à un nombre fini d'états. *Revue Mathématique de l'Union Interbalkanique*, 2:77–105, 1938.

[Dud68]  Richard M. Dudley. Distances of probability measures and random variables. *The Annals of Mathematical Statistics*, 39(5):1563–1572, October 1968.

[DW12]  Constantinos Daskalakis and Seth Matthew Weinberg. Symmetries and optimal multidimensional mechanism design. In *Proceedings of the 13th ACM Conference on Electronic Commerce*, EC '12, page 370–387, New York, NY, USA, 2012. Association for Computing Machinery.

[FGL14]  Michal Feldman, Nick Gravin, and Brendan Lucier. Combinatorial auctions via posted prices. In *Proceedings of the twenty-sixth annual ACM-SIAM symposium on Discrete algorithms*, pages 123–135. SIAM, 2014.

[GHZ19]   Chenghao Guo, Zhiyi Huang, and Xinzhi Zhang. Settling the sample complexity of single-parameter revenue maximization. In *Proceedings of the 51st Annual ACM SIGACT Symposium on Theory of Computing*, pages 662–673, 2019.

[GL18]   Nick Gravin and Pinyan Lu. Separation in correlation-robust monopolist problem with budget. In *Proceedings of the Twenty-Ninth Annual ACM-SIAM Symposium on Discrete Algorithms*, pages 2069–2080. SIAM, 2018.

[GPTD23]   Yiannis Giannakopoulos, Diogo Poças, and Alexandros Tsigonias-Dimitriadis. Robust revenue maximization under minimal statistical information. *ACM Transactions on Economics and Computation*, 10(3):1–34, 2023.

[Gri75]   David Griffeath. A maximal coupling for markov chains. *Zeitschrift für Wahrscheinlichkeitstheorie und Verwandte Gebiete*, 31:95–106, June 1975.

[GW21]   Yannai A Gonczarowski and S Matthew Weinberg. The sample complexity of up-to-$\varepsilon$ multi-dimensional revenue maximization. *Journal of the ACM (JACM)*, 68(3):1–28, 2021.

[HJW15]   Yanjun Han, Jiantao Jiao, and Tsachy Weissman. Minimax estimation of discrete distributions under $\ell_1$ loss. *IEEE Transactions on Information Theory*, 61(11):6343–6354, November 2015.

[HMR15]   Zhiyi Huang, Yishay Mansour, and Tim Roughgarden. Making the most of your samples. In *Proceedings of the Sixteenth ACM Conference on Economics and Computation*, pages 45–60, 2015.

[HN13]   Sergiu Hart and Noam Nisan. The menu-size complexity of auctions. *ACM Conference on Electronic Commerce*, 04 2013.

[HN19]   Sergiu Hart and Noam Nisan. Selling multiple correlated goods: Revenue maximization and menu-size complexity. *Journal of Economic Theory*, 183:991–1029, 2019.

[Kal21]   Olav Kallenberg. *Foundations of Modern Probability*, volume 99 of *Probability Theory and Stochastic Modelling*. Springer, New York, NY, USA, third edition, 2021.

[Kan60]   Leonid V. Kantorovich. Mathematical methods of organizing and planning production. *Management Science*, 6(4):366–422, July 1960.

[KS80]   Ross Kindermann and Laurie Snell. *Markov random fields and their applications*, volume 1. American Mathematical Society, 1980.

[KW19]   Robert Kleinberg and S Matthew Weinberg. Matroid prophet inequalities and applications to multi-dimensional mechanism design. *Games and Economic Behavior*, 113:97–115, 2019.

[LLY19]   Yingkai Li, Pinyan Lu, and Haoran Ye. Revenue maximization with imprecise distribution. In *Proceedings of the 18th International Conference on Autonomous Agents and MultiAgent Systems*, pages 1582–1590, 2019.

[LPW09]   David A. Levin, Yuval Peres, and Elizabeth L. Wilmer. *Markov Chains and Mixing Times*. American Mathematical Society, Providence, RI, USA, first edition, 2009.

[Luc17]   Brendan Lucier. An economic view of prophet inequalities. *ACM SIGecom Exchanges*, 16(1):24–47, 2017.

[LY13]   Xinye Li and Andrew Chi-Chih Yao. On revenue maximization for selling multiple independently distributed items. *Proceedings of the National Academy of Sciences*, 110(28):11232–11237, 2013.

[Mak19]   Anuran Makur. *Information Contraction and Decomposition*. Sc.D. thesis in Electrical Engineering and Computer Science, Massachusetts Institute of Technology, Cambridge, MA, USA, May 2019.

[MR16]    Jamie Morgenstern and Tim Roughgarden. Learning simple auctions. In *Conference on Learning Theory*, pages 1298–1318. PMLR, 2016.

[PSCW22]  Alexandros Psomas, Ariel Schvartzman Cohenca, and S Weinberg. On infinite separations between simple and optimal mechanisms. *Advances in Neural Information Processing Systems*, 35:4818–4829, 2022.

[PSW19]   Alexandros Psomas, Ariel Schvartzman, and S Matthew Weinberg. Smoothed analysis of multi-item auctions with correlated values. In *Proceedings of the 2019 ACM Conference on Economics and Computation*, pages 417–418. ACM, 2019.

[PW22]    Yury Polyanskiy and Yihong Wu. *Information Theory: From Coding to Learning*. Cambridge University Press Preprint, New York, NY, USA, 2022.

[RS17]    Aviad Rubinstein and Sahil Singla. Combinatorial prophet inequalities. In *Proceedings of the Twenty-Eighth Annual ACM-SIAM Symposium on Discrete Algorithms*, pages 1671–1687. SIAM, 2017.

[RW15]    Aviad Rubinstein and S Matthew Weinberg. Simple mechanisms for a subadditive buyer and applications to revenue monotonicity. In *Proceedings of the Sixteenth ACM Conference on Economics and Computation*, pages 377–394. ACM, 2015.

[RW18]    Aviad Rubinstein and S. Matthew Weinberg. Simple mechanisms for a subadditive buyer and applications to revenue monotonicity. *ACM Trans. Econ. Comput.*, 6(3–4), oct 2018.

[SC84]    Ester Samuel-Cahn. Comparison of Threshold Stop Rules and Maximum for Independent Nonnegative Random Variables. *The Annals of Probability*, 12(4):1213 – 1216, 1984.

[SK75]    David Sherrington and Scott Kirkpatrick. Solvable model of a spin-glass. *Phys. Rev. Lett.*, 35:1792–1796, Dec 1975.

[Sko56]   Anatoliy V. Skorokhod. Limit theorems for stochastic processes. *Theory of Probability and Its Applications*, 1(3):261–290, 1956.

[ST04]    Daniel A Spielman and Shang-Hua Teng. Nearly-linear time algorithms for graph partitioning, graph sparsification, and solving linear systems. In *Proceedings of the thirty-sixth annual ACM symposium on Theory of computing*, pages 81–90. ACM, 2004.

[Str65]   Volker Strassen. The existence of probability measures with given marginals. *The Annals of Mathematical Statistics*, 36(2):423–439, April 1965.

[Tsy08]   Alexandre B. Tsybakov. *Introduction to Nonparametric Estimation*. Springer Publishing Company, Incorporated, 1st edition, 2008.

[Vil09]   Cédric Villani. *Optimal Transport: Old and New*, volume 338 of *Grundlehren der mathematischen Wissenschaften*. Springer, Berlin, Heidelberg, Germany, 2009.

[Wic70]   Michael J. Wichura. On the construction of almost uniformly convergent random variables with given weakly convergent image laws. *The Annals of Mathematical Statistics*, 4141(1):284–291, February 1970.

[Yao15]   Andrew Chi-Chih Yao. An n-to-1 bidder reduction for multi-item auctions and its applications. In *Proceedings of the Twenty-Sixth Annual ACM-SIAM Symposium on Discrete Algorithms*, pages 92–109. Society for Industrial and Applied Mathematics, 2015.

# A   Proofs missing from Section 3

The following simple proposition will also be useful in multiple proofs throughout this appendix.

**Proposition 5.** *Let $\mathcal{M}$ be an ex-post IR mechanism. Then, $-H \leq u_i^{\mathcal{M}}(t_i \leftarrow t_i', t_{-i}) \leq 3H$, for all $i \in [n], t_i, t_i' \in \mathcal{T}_i, t_{-i} \in \mathcal{T}_{-i}$.*

*Proof of Proposition 5.* Since $\mathcal{M}$ is ex-post IR, we have that $t_i\left(\mathcal{M}(t_i, t_{-i})\right) \geq 0$, for all $i \in [n], t_i \in \mathcal{T}_i, t_{-i} \in \mathcal{T}_{-i}$. Furthermore, since payments are lower bounded by $-H$, and since the valuations are bounded and quasi-linear, we have that $t_i\left(\mathcal{M}(t_i', t_{-i})\right) \leq 2H$, for all $i \in [n], t_i, t_i' \in \mathcal{T}_i, t_{-i} \in \mathcal{T}_{-i}$. Since payments are also upper bounded by $H$ (due to the ex-post IR constraint), and valuations are non-negative, we also have $t_i\left(\mathcal{M}(t_i', t_{-i})\right) \geq -H$, for all $i \in [n], t_i, t_i' \in \mathcal{T}_i, t_{-i} \in \mathcal{T}_{-i}$. Combining these inequalities we have $-H \leq u_i(t_i \leftarrow t_i', t_{-i}) \leq 3H$, for all $i \in [n], t_i, t_i' \in \mathcal{T}_i, t_{-i} \in \mathcal{T}_{-i}$.   $\square$

## A.1   Relaxing the assumptions in Theorem 1

We start by showing that, in sharp contrast to BIC, the DSIC property is much easier to "propagate" from a small set of types to a larger set, using the following construction.

**Definition 3** (DSIC extension of a mechanism). *Let $\mathcal{T}_i^+ \subseteq \mathcal{T}_i$ be a subset of possible types for agent $i \in [n]$, such that $\perp \in \mathcal{T}_i^+$, and let $\mathcal{M} = (x, p)$ be a mechanism defined on types $\times_{i \in [n]} \mathcal{T}_i^+$. The extension of $\mathcal{M}$ to $\mathcal{T}$ is the mechanism $\widehat{\mathcal{M}} = (\widehat{x}, \widehat{p})$, where for reported types $t = (t_1, \cdots, t_n)$:*

1. *If $\times_{i \in [n]} \mathcal{T}_i^+$, then $\widehat{x}(t) = x(t)$ and $\widehat{p}(t) = \widehat{p}(t)$.*

2. *If there exists $i$, such that $t_i \notin \mathcal{T}_i^+$ and $\forall j \in [n]/\{i\} : t_j \in \mathcal{T}_j^+$ then $\widehat{x}_i(t) = x_i(t_i', t_{-i})$ and $\widehat{p}_i(t) = \widehat{p}_i(t_i', t_{-i})$, where $t_i' = \arg\max_{z_i \in \mathcal{T}_i^+} t_i(\mathcal{M}(z_i, t_{-i}))$. For each $j \in [n]/\{i\}$ we have that $\widehat{x}_j(t) = 0$ and $\widehat{p}_j(t) = 0$ (They receive nothing, and pay nothing).*

3. *If there exist $i, i'$ such that $i \neq i'$ and $t_i \notin \mathcal{T}_i^+$ and $t_{i'} \notin \mathcal{T}_{i'}^+$, then nobody receives and pays nothing (i.e. $x(t) = 0, \widehat{p}(t) = 0$).*

A similar construction appears in [DFK11], in the context of implementing the solution of a linear program as a DSIC auction.

**Lemma 6.** *Let $\mathcal{T}_i^+ \subseteq \mathcal{T}_i$ be a subset of possible types for agent $i \in [n]$, such that $\perp \in \mathcal{T}_i^+$, and let $\mathcal{M} = (x, p)$ be a DSIC and ex-post IR mechanism defined on types $\mathcal{T}^+ = \times_{i \in [n]} \mathcal{T}_i^+$. Then, the extension of $\mathcal{M}$ to $\mathcal{T}$, $\widehat{\mathcal{M}} = (\widehat{x}, \widehat{p})$, is DSIC and ex-post IR.*

*Proof of Lemma 6.* The fact that $\widehat{\mathcal{M}}$ is ex-post IR is trivial for cases 1 and 3 of Definition 3. For case 2, it is trivial that it is ex-post IR for all $j \in [n]/\{i\}$. Also since $\perp \in \mathcal{T}_i^+$ we have that $\max_{z_i \in \mathcal{T}_i^+} t_i(\mathcal{M}(z_i, t_{-i})) \geq t_i(\mathcal{M}(\perp, t_{-i})) \geq 0$, which implies that the mechanism is ex-post IR for agent $i$.

Next, we argue that $\widehat{\mathcal{M}}$ is DSIC. If $t \in \mathcal{T}^+$, then any misreport $t_i'$ of agent $i$ will also get mapped to a type in $\mathcal{T}_i^+$; since $\mathcal{M}$ is DSIC, agent $i$ cannot increase her utility by deviating. If $t$ falls into the second case, an agent $j \in [n]/\{i\}$ receives nothing and pays nothing, no matter what she reports. If agent $i$ misreports a type $t_i'$, she either receives utility $t_i(\mathcal{M}(t_i', t_{-i}))$, if $t_i' \in \mathcal{T}_i^+$, or $t_i(\mathcal{M}((t^*)', t_{-i}))$, where $(t^*)' = \arg\max_{z_i \in \mathcal{T}_i^+} t_i'(\mathcal{M}(z_i, t_{-i}))$, if $t_i' \notin \mathcal{T}_i^+$, both of which are (weakly) worse than $\max_{z_i \in \mathcal{T}_i^+} t_i(\mathcal{M}(z_i, t_{-i}))$, her utility when reporting $t_i$. Finally, in case 3, every agent $i$ always receives nothing and pays nothing, even after unilaterally changing her report.   $\square$

Thus without loss of generality, we can always assume that DSIC mechanism defined on a subset of the type space $\mathcal{T}^+ \subseteq \mathcal{T}$ is DSIC on all bids in $\mathcal{T}$.

## A.2 Proofs missing from Section 3.2

*Proof of Lemma 3.*

$$2\,d_{\mathsf{TV}}\left(P_{X,Y},Q_{X,Y}\right) = \sum_x \sum_y |P_{X,Y}(x,y) - Q_{X,Y}(x,y)|$$

$$\geq \sum_{x:Q_X(x)>0} \sum_y |P_{X,Y}(x,y) - Q_{X,Y}(x,y)|$$

$$= \sum_{x:Q_X(x)>0} Q_X(x) \sum_y \left| P_{Y|X=x}(y)\frac{P_X(x)}{Q_X(x)} - Q_{Y|X=x}(y) - P_{Y|X=x}(y) + P_{Y|X=x}(y) \right|$$

$$\geq \sum_{x:Q_X(x)>0} Q_X(x) \sum_y \left( |P_{Y|X=x}(y) - Q_{Y|X=x}(y)| - P_{Y|X=x}(y)\left|1 - \frac{P_X(x)}{Q_X(x)}\right| \right)$$

$$= \sum_{x:Q_X(x)>0} Q_X(x) \left( 2\,d_{\mathsf{TV}}\left(P_{Y|X=x},Q_{Y|X=x}\right) - \frac{|Q_X(x) - P_X(x)|}{Q_X(x)} \right)$$

$$\geq \left( 2\sum_x Q_X(x)\,d_{\mathsf{TV}}\left(P_{Y|X=x},Q_{Y|X=x}\right) \right) - 2\,d_{\mathsf{TV}}\left(Q_X,P_X\right).$$

Re-arranging, we have that

$$\mathbb{E}_{x\sim Q_X}\left[d_{\mathsf{TV}}\left(P_{Y|X=x},Q_{Y|X=x}\right)\right] \leq d_{\mathsf{TV}}\left(P_{X,Y},Q_{X,Y}\right) + d_{\mathsf{TV}}\left(Q_X,P_X\right).$$

The data processing inequality gives us that $d_{\mathsf{TV}}\left(Q_X,P_X\right) \leq d_{\mathsf{TV}}\left(P_{X,Y},Q_{X,Y}\right)$ [PW22, Theorem 7.4], and thus we have $\mathbb{E}_{x\sim Q_X}\left[d_{\mathsf{TV}}\left(P_{Y|X=x},Q_{Y|X=x}\right)\right] \leq 2\,d_{\mathsf{TV}}\left(P_{X,Y},Q_{X,Y}\right)$, as desired. For distributions supported over continuous sets, the proof follows with similar arguments.

So far, we have established that $\mathbb{E}_{x\sim Q_X}\left[d_{\mathsf{TV}}\left(P_{Y|X=x},Q_{Y|X=x}\right)\right] \leq d_{\mathsf{TV}}\left(P_{X,Y},Q_{X,Y}\right) + d_{\mathsf{TV}}\left(Q_X,P_X\right)$. Using Markov's inequality completes the proof of Lemma 3. □

*Proof of Lemma 4.* $\mathcal{M}$ is ex-post IR for $\mathcal{D}'$, by definition. Let $\mathcal{D}_{-i|t_i}$ be the probability distribution for the valuations of every agent except $i$, conditioned on the event that the type of agent $i$ is $t_i \in \mathcal{T}_i$. Proposition 5 implies that $u_i^{\mathcal{M}}(t_i \leftarrow w_i, t_{-i}) \in [-H, 3H]$, for all $i \in [n], t_i, w_i \in \mathcal{T}_i, t_{-i} \in \mathcal{T}_{-i}$, and therefore $u_i^{\mathcal{M}}(t_i \leftarrow w_i, t_{-i}) - u_i^{\mathcal{M}}(t_i \leftarrow w_i, t'_{-i}) \leq 4H\,\mathbb{1}\{t_{-i} \neq t'_{-i}\}$. Thus, for any coupling $\gamma$ of $\mathcal{D}_{-i|t_i}$ and $\mathcal{D}'_{-i|t_i}$, and specifically for the optimal coupling $\gamma^*$ between $\mathcal{D}_{-i|t_i}$ and $\mathcal{D}'_{-i|t_i}$ (see Definition 2), we have:

$$\mathbb{E}_{(t_{-i},t'_{-i})\sim\gamma^*}\left[u_i^{\mathcal{M}}(t_i \leftarrow w_i, t_{-i}) - u_i^{\mathcal{M}}(t_i \leftarrow w_i, t'_{-i})\right] \leq 4H\,\mathbb{E}_{(t_{-i},t'_{-i})\sim\gamma^*}\left[\mathbb{1}\{t_{-i} \neq t'_{-i}\}\right]$$

$$\leq 4H\,d_{\mathsf{TV}}\left(\mathcal{D}_{-i|t_i},\mathcal{D}'_{-i|t_i}\right).$$

Using linearity of expectation and re-arranging we have:

$$-\mathbb{E}_{t'_{-i}\sim\mathcal{D}'_{-i}|t_i}\left[u_i^{\mathcal{M}}(t_i \leftarrow w_i, t'_{-i})\right] \leq 4H\,d_{\mathsf{TV}}\left(\mathcal{D}_{-i|t_i},\mathcal{D}'_{-i|t_i}\right) - \mathbb{E}_{t_{-i}\sim\mathcal{D}_{-i}|t_i}\left[u_i^{\mathcal{M}}(t_i \leftarrow w_i, t_{-i})\right].$$

By setting $Q_X = \mathcal{D}'_i$, $P_{Y|X=x} = \mathcal{D}_{-i|t_i}$, and $Q_{Y|X=x} = \mathcal{D}'_{-i|t_i}$ in Lemma 3 we have that, with probability at least $1-q$, $d_{\mathsf{TV}}\left(\mathcal{D}_{-i|t_i},\mathcal{D}'_{-i|t_i}\right) \leq \frac{2}{q}\,d_{\mathsf{TV}}\left(\mathcal{D},\mathcal{D}'\right) \leq 2\frac{\delta}{q}$. Therefore, with probability at least $1-q$:

$$-\mathbb{E}_{t'_{-i}\sim\mathcal{D}'_{-i}|t_i}\left[u_i^{\mathcal{M}}(t_i \leftarrow w_i, t'_{-i})\right] \leq 4H\,d_{\mathsf{TV}}\left(\mathcal{D}_{-i|t_i},\mathcal{D}'_{-i|t_i}\right) - \mathbb{E}_{t_{-i}\sim\mathcal{D}_{-i}|t_i}\left[u_i^{\mathcal{M}}(t_i \leftarrow w_i, t_{-i})\right]$$

$$\leq 8H\frac{\delta}{q} - \mathbb{E}_{t_{-i}\sim\mathcal{D}_{-i}|t_i}\left[u_i^{\mathcal{M}}(t_i \leftarrow w_i, t_{-i})\right]$$

$$\leq \frac{8H\delta}{q},$$

where the last inequality uses the fact that $\mathcal{M}$ is BIC. Replacing with the definition of $u_i^{\mathcal{M}}(t_i \leftarrow w_i, t'_{-i})$ we get $-\mathbb{E}_{t_{-i}\sim\mathcal{D}'_{-i}|t_i}\left[t_i\left(\mathcal{M}(t_i, t_{-i})\right)\right] + \mathbb{E}_{t_{-i}\sim\mathcal{D}'_{-i}|t_i}\left[t_i\left(\mathcal{M}(w_i, t_{-i})\right)\right] \leq \frac{8H\delta}{q}$, with probability at least $1-q$. Re-arranging we get the desired $(\varepsilon, q)$ BIC constraint. □

# B  Proofs missing from Section 4.1

In order to prove Lemma 5, it will be convenient to define the following notion of an extension of a BIC mechanism.

**Definition 4** (BIC extension of a mechanism). *Let $\mathcal{T}_i^+ \subseteq \mathcal{T}_i$ be a subset of types for agent $i \in [n]$ such that $\perp \in \mathcal{T}_i^+$, and let $\mathcal{M} = (x, p)$ be a mechanism defined on types in $\times_{i \in [n]} \mathcal{T}_i^+$. Let $\mathcal{T}_i^- = \mathcal{T}_i - \mathcal{T}_i^+$, and consider the mapping*

$$\tau_i(t_i) = \begin{cases} t_i, & \text{if } t_i \in \mathcal{T}_i^+ \\ \arg\max_{z \in \mathcal{T}_i^+} \mathbb{E}_{t_{-i} \sim \mathcal{D}_{-i}} \left[ t_i(\mathcal{M}(z, t_{-i})) \right], & \text{if } t_i \in \mathcal{T}_i^- \end{cases}$$

*The extension of $\mathcal{M}$ to $\mathcal{T}$ is the mechanism $\widehat{\mathcal{M}} = (\widehat{x}, \widehat{p})$, where $\widehat{x}(t) = x(\tau(t))$, and for all $i \in [n]$,*

$$\widehat{p}_i(t_i, t_{-i}) = \begin{cases} p_i(t_i, t_{-i}), & \text{if } t_i \in \mathcal{T}_i^+ \\ v_i(\widehat{x}(t_i, t_{-i})) \frac{\mathbb{E}_{t_{-i} \sim \mathcal{D}_{-i}}[p_i(\tau_i(t_i), t_{-i}]}{\mathbb{E}_{t_{-i} \sim \mathcal{D}_{-i}}[v_i(x(\tau_i(t_i), t_{-i}))]}, & \text{if } t_i \in \mathcal{T}_i^- \end{cases}$$

We prove the following technical lemma.

**Lemma 7.** *Let $\mathcal{T}_i^+ \subseteq \mathcal{T}_i$ be a subset of types for agent $i \in [n]$ such that $\perp \in \mathcal{T}_i^+$, and let $\mathcal{D} = \times_{i \in [n]} \mathcal{D}_i$ be a product distribution, where each $\mathcal{D}_i$ is supported on $\mathcal{T}_i$. Let $\mathcal{M} = (x, p)$ be an ex-post IR mechanism which satisfies $\mathbb{E}_{t_{-i} \sim \mathcal{D}_{-i}} \left[ u_i^{\mathcal{M}}(t_i \leftarrow w_i, t_{-i}) \right] \geq -\varepsilon$, for all $t_i \in \mathcal{T}_i^+, w_i \in \mathcal{T}_i$.*

*Then, for any product distribution $\widehat{\mathcal{D}} = \times_{i \in [n]} \widehat{\mathcal{D}}_i$ such that $d_{\mathsf{TV}}\left(\mathcal{D}, \widehat{\mathcal{D}}\right) \leq \delta$, the extension of $\mathcal{M}$ to $\mathcal{T}$ (as defined in Definition 4) is ex-post IR and $O\left(\varepsilon + (\beta n + \delta) H\right)$-BIC with respect to $\widehat{\mathcal{D}}$, where $\beta = 1 - \Pr_{t_i \sim \widehat{\mathcal{D}}_i} \left[ t_i \in \mathcal{T}_i^+ \right]$. Furthermore, $Rev(\widehat{\mathcal{M}}, \widehat{\mathcal{D}}) \geq Rev(\mathcal{M}, \mathcal{D}) - V(\beta n + \delta)$.*

*Proof of Lemma 7.* Let $\widehat{\mathcal{M}} = (\widehat{x}, \widehat{p})$ be the extension of $\mathcal{M}$ to $\mathcal{T}$. First, we argue that $\widehat{\mathcal{M}}$ is ex-post IR. Since $\mathcal{M}$ is ex-post IR, the ex-post IR condition for $\widehat{\mathcal{M}}$ is satisfied for all $t_i \in \mathcal{T}_i^+$, by construction. For a type $t_i \in \mathcal{T}_i^-$, since $\perp \in \mathcal{T}_i^+$ and $\tau_i(t_i) \in \mathcal{T}_i^+$, we have that $\mathbb{E}_{t_{-i} \sim \mathcal{D}_{-i}}[t_i(\mathcal{M}(\tau_i(t_i), t_{-i}))] \geq \mathbb{E}_{t_{-i} \sim \mathcal{D}_{-i}}[t_i(\mathcal{M}(\perp, t_{-i}))] = 0$. Therefore, $\mathbb{E}_{t_{-i} \sim \mathcal{D}_{-i}}[p_i(\tau_i(t_i), t_{-i})] \leq \mathbb{E}_{t_{-i} \sim \mathcal{D}_{-i}}[v_i(x(\tau_i(t_i), t_{-i}))]$, which implies that $v_i(\widehat{x}(t)) - \widehat{p}_i(t) = v_i(\widehat{x}(t)) - v_i(\widehat{x}(t)) \frac{\mathbb{E}_{t_{-i} \sim \mathcal{D}_{-i}}[p_i(\tau_i(t_i), t_{-i}]}{\mathbb{E}_{t_{-i} \sim \mathcal{D}_{-i}}[v_i(x(\tau_i(t_i), t_{-i}))]} \geq 0$.

Next, we prove the BIC guarantee of $\widehat{\mathcal{M}}$. Towards this, first define $\tau(\widehat{\mathcal{D}})$ as the distribution induced by first sampling from $\widehat{\mathcal{D}}$, and then apply mapping $\tau(.)$, as defined in Definition 4. The tensorization property of TV distance [LPW09, Chapter 4] implies that $d_{\mathsf{TV}}\left(\widehat{\mathcal{D}}, \tau(\widehat{\mathcal{D}})\right) \leq \beta n$, and thus from the triangle inequality, $d_{\mathsf{TV}}\left(\mathcal{D}, \tau(\widehat{\mathcal{D}})\right) \leq \delta + \beta n$. Our goal is to prove the following lower bound:

$$\mathbb{E}_{t_{-i} \sim \widehat{\mathcal{D}}_{-i}} \left[ u_i^{\widehat{\mathcal{M}}}(t_i \leftarrow w_i, t_{-i}) \right] \geq -\left( 4\left(\frac{3}{2}\delta + \beta n\right) H + 4\delta H + \varepsilon \right).$$

We first prove the following intermediate bound:

$$\mathbb{E}_{t_{-i} \sim \widehat{\mathcal{D}}_{-i}} \left[ u_i^{\widehat{\mathcal{M}}}(t_i \leftarrow w_i, t_{-i}) \right] \geq \mathbb{E}_{t_{-i} \sim \widehat{\mathcal{D}}_{-i}} \left[ u_i^{\mathcal{M}}(\tau(t_i) \leftarrow \tau(w_i), t_{-i}) \right] - 4\left(\frac{3}{2}\delta + \beta n\right) H$$

Generally, our bounds will be trivial when $t_i \in \mathcal{T}_i^+$ due to the nature of $\widehat{\mathcal{M}}$. So the main focus of the analysis is to prove those bounds for $t_i \in \mathcal{T}_i^-$.

First, we prove two inequalities that will be useful in our analysis.

$$\mathbb{E}_{t_{-i} \sim \mathcal{D}_{-i}} \left[ v_i(x_i(\tau(t_i), t_{-i})) \right] \leq \mathbb{E}_{t_{-i} \sim \mathcal{D}_{-i}} \left[ \widehat{x}_i(t_i, t_{-i}) \right] + H \beta n. \tag{2}$$

$$\mathbb{E}_{t_{-i} \sim \mathcal{D}_{-i}} \left[ p_i(\tau(t_i), t_{-i}) \right] \geq \mathbb{E}_{t_{-i} \sim \mathcal{D}_{-i}} \left[ \widehat{p}_i(t_i, t_{-i}) \right] - H \beta n. \tag{3}$$

For inequality (2), using Lemma 2 we can get:

$$\mathbb{E}_{t_{-i}\sim\mathcal{D}_{-i}}\left[v_i(x_i(\tau(t_i),t_{-i}))\right] \le \mathbb{E}_{t_{-i}\sim\tau(\mathcal{D}_{-i})}\left[v_i(x_i(\tau(t_i),t_{-i}))\right] + H\ d_{\mathsf{TV}}\left(\mathcal{D}_{-i},\tau(\mathcal{D}_{-i})\right)$$
$$\le \mathbb{E}_{t_{-i}\sim\tau(\mathcal{D}_{-i})}\left[v_i(x_i(\tau(t_i),t_{-i}))\right] + H\ d_{\mathsf{TV}}\left(\mathcal{D},\tau(\mathcal{D})\right)$$
$$\le \mathbb{E}_{t_{-i}\sim\tau(\mathcal{D}_{-i})}\left[v_i(x_i(\tau(t_i),t_{-i}))\right] + H\ \beta n$$
$$\le \mathbb{E}_{t_{-i}\sim\mathcal{D}_{-i}}\left[x_i(\tau(t_i),\tau(t_{-i}))\right] + H\ \beta n$$
$$\le \mathbb{E}_{t_{-i}\sim\mathcal{D}_{-i}}\left[\widehat{x}_i(t_i,t_{-i})\right] + H\ \beta n.$$

Similarly, for inequality (3):

$$\mathbb{E}_{t_{-i}\sim\mathcal{D}_{-i}}\left[p_i(\tau(t_i),t_{-i})\right] = \mathbb{E}_{t_{-i}\sim\mathcal{D}_{-i}}\left[p_i(\tau(t_i),t_{-i})\right]\frac{\mathbb{E}_{t'_{-i}\sim\mathcal{D}_{-i}}\left[v_i(x_i(\tau(t_i),t'_{-i}))\right]}{\mathbb{E}_{t_{-i}\sim\mathcal{D}_{-i}}\left[v_i(x_i(\tau(t_i),t_{-i}))\right]}$$

$$= \mathbb{E}_{t'_{-i}\sim\mathcal{D}_{-i}}\left[v_i(x_i(\tau(t_i),t'_{-i}))\frac{\mathbb{E}_{t_{-i}\sim\mathcal{D}_{-i}}\left[p_i(\tau(t_i),t_{-i})\right]}{\mathbb{E}_{t_{-i}\sim\mathcal{D}_{-i}}\left[v_i(x_i(\tau(t_i),t_{-i}))\right]}\right].$$

We've already shown, when arguing the ex-post IR property, that $\frac{\mathbb{E}_{t_{-i}\sim\mathcal{D}_{-i}}[p_i(\tau(t_i),t_{-i})]}{\mathbb{E}_{t_{-i}\sim\mathcal{D}_{-i}}[v_i(x_i(\tau(t_i),t_{-i}))]} \le 1$ and thus $v_i(x_i(\tau(t_i),t'_{-i}))\frac{\mathbb{E}_{t_{-i}\sim\mathcal{D}_{-i}}[p_i(\tau(t_i),t_{-i})]}{\mathbb{E}_{t_{-i}\sim\mathcal{D}_{-i}}[v_i(x_i(\tau(t_i),t_{-i}))]} \in [0,H]$. Therefore, we can use Lemma 2 for $\mathcal{D}_{-i}$ and $\tau(\mathcal{D}_{-i})$ on this function (as the objective) to get:

$$\mathbb{E}_{t_{-i}\sim\mathcal{D}_{-i}}\left[p_i(\tau(t_i),t_{-i})\right] = \mathbb{E}_{t'_{-i}\sim\mathcal{D}_{-i}}\left[v_i(x_i(\tau(t_i),t'_{-i}))\frac{\mathbb{E}_{t_{-i}\sim\mathcal{D}_{-i}}\left[p_i(\tau(t_i),t_{-i})\right]}{\mathbb{E}_{t_{-i}\sim\mathcal{D}_{-i}}\left[v_i(x_i(\tau(t_i),t_{-i}))\right]}\right]$$

$$\ge \mathbb{E}_{t'_{-i}\sim\tau(\mathcal{D}_{-i})}\left[v_i(x_i(\tau(t_i),t'_{-i}))\frac{\mathbb{E}_{t_{-i}\sim\mathcal{D}_{-i}}\left[p_i(\tau(t_i),t_{-i})\right]}{\mathbb{E}_{t_{-i}\sim\mathcal{D}_{-i}}\left[v_i(x_i(\tau(t_i),t_{-i}))\right]}\right] - H\ d_{\mathsf{TV}}\left(\mathcal{D}_{-i},\tau(\mathcal{D}_{-i})\right)$$

$$\ge \mathbb{E}_{t'_{-i}\sim\tau(\mathcal{D}_{-i})}\left[v_i(x_i(\tau(t_i),t'_{-i}))\frac{\mathbb{E}_{t_{-i}\sim\mathcal{D}_{-i}}\left[p_i(\tau(t_i),t_{-i})\right]}{\mathbb{E}_{t_{-i}\sim\mathcal{D}_{-i}}\left[v_i(x_i(\tau(t_i),t_{-i}))\right]}\right] - H\ d_{\mathsf{TV}}\left(\mathcal{D},\tau(\mathcal{D})\right)$$

$$\ge \mathbb{E}_{t'_{-i}\sim\tau(\mathcal{D}_{-i})}\left[v_i(x_i(\tau(t_i),t'_{-i}))\frac{\mathbb{E}_{t_{-i}\sim\mathcal{D}_{-i}}\left[p_i(\tau(t_i),t_{-i})\right]}{\mathbb{E}_{t_{-i}\sim\mathcal{D}_{-i}}\left[v_i(x_i(\tau(t_i),t_{-i}))\right]}\right] - H\ \beta n$$

$$= \mathbb{E}_{t'_{-i}\sim\mathcal{D}_{-i}}\left[v_i(x_i(\tau(t_i),\tau(t'_{-i})))\frac{\mathbb{E}_{t_{-i}\sim\mathcal{D}_{-i}}\left[p_i(\tau(t_i),t_{-i})\right]}{\mathbb{E}_{t_{-i}\sim\mathcal{D}_{-i}}\left[v_i(x_i(\tau(t_i),t_{-i}))\right]}\right] - H\ \beta n$$

$$= \mathbb{E}_{t'_{-i}\sim\mathcal{D}_{-i}}\left[\widehat{p}_i(t_i,t'_{-i})\right] - H\ \beta n.$$

With inequalities (2) and (3) at hand, we are ready to show the following, for all $t_i \in \mathcal{T}_i^-$:

$$\mathbb{E}_{t_{-i}\sim\widehat{\mathcal{D}}_{-i}}\left[t_i\left(\mathcal{M}(\tau(t_i),t_{-i})\right)\right] \le^{(Lemma\ 2)} \mathbb{E}_{t_{-i}\sim\mathcal{D}_{-i}}\left[t_i\left(\mathcal{M}(\tau(t_i),t_{-i})\right)\right] + 2\delta H$$

$$= \mathbb{E}_{t_{-i}\sim\mathcal{D}_{-i}}\left[(v_i(x_i(\tau(t_i),t_{-i})) - p_i(\tau(t_i),t_{-i}))\right] + 2\delta H$$

$$\le^{(Ineq.\ (2)\ and\ (3))} \mathbb{E}_{t_{-i}\sim\mathcal{D}_{-i}}\left[\widehat{x}_i(t_i,t_{-i})\right] - \mathbb{E}_{t_{-i}\sim\mathcal{D}_{-i}}\left[\widehat{p}_i(t_i,t_{-i})\right] + 2(\delta + \beta n)\,H$$

$$= \mathbb{E}_{t_{-i}\sim\mathcal{D}_{-i}}\left[t_i\left(\widehat{\mathcal{M}}(t_i,t_{-i})\right)\right] + 2(\delta + \beta n)\,H$$

$$\le^{(Lemma\ 2)} \mathbb{E}_{t_{-i}\sim\widehat{\mathcal{D}}_{-i}}\left[t_i\left(\widehat{\mathcal{M}}(t_i,t_{-i})\right)\right] + 2\left(\frac{3}{2}\delta + \beta n\right)H.$$

Whenever $t_i \in \mathcal{T}_i^+$ we can directly argue that:

$$\mathbb{E}_{y_{-i}\sim\widehat{\mathcal{D}}_{-i}}\left[t_i\left(\mathcal{M}(\tau(t_i),t_{-i})\right)\right] \le \mathbb{E}_{t_{-i}\sim\tau(\widehat{\mathcal{D}})_{-i}}\left[t_i\left(\mathcal{M}(\tau(t_i),t_{-i})\right)\right] + \beta n H$$

$$= \mathbb{E}_{t_{-i}\sim\widehat{\mathcal{D}}_{-i}}\left[t_i\left(\mathcal{M}(\tau(t_i),\tau(t_{-i}))\right)\right] + \beta n H$$

$$= \mathbb{E}_{t_{-i}\sim\widehat{\mathcal{D}}_{-i}}\left[t_i\left(\widehat{\mathcal{M}}(t_i,t_{-i})\right)\right] + \beta n H.$$

Similarly, we get that $\mathbb{E}_{t_{-i}\sim\widehat{\mathcal{D}}_{-i}}\left[t_i\left(\mathcal{M}(\tau(w_i),t_{-i})\right)\right] \geq \mathbb{E}_{t_{-i}\sim\widehat{\mathcal{D}}_{-i}}\left[t_i\left(\widehat{\mathcal{M}}(w_i,t_{-i})\right)\right] - 2(\frac{3}{2}\delta + \beta n)\,H$ for all $w_i \in \mathcal{T}_i$. Combining we get that for $t_i \in \mathcal{T}_i^-, w_i \in \mathcal{T}_i$:

$$\mathbb{E}_{t_{-i}\sim\widehat{\mathcal{D}}_{-i}}\left[t_i(\widehat{\mathcal{M}}(t_i,t_{-i})\right] - \mathbb{E}_{t_{-i}\sim\widehat{\mathcal{D}}_{-i}}\left[t_i(\widehat{\mathcal{M}}(w_i,t_{-i})\right] \geq$$

$$\mathbb{E}_{t_{-i}\sim\widehat{\mathcal{D}}_{-i}}\left[t_i(\mathcal{M}(\tau(t_i),t_{-i})\right] - \mathbb{E}_{t_{-i}\sim\widehat{\mathcal{D}}_{-i}}\left[t_i(\mathcal{M}(\tau(w_i),t_{-i})\right] - 4\left(\frac{3}{2}\delta + \beta n\right)H,$$

and for $t_i \in \mathcal{T}_i^+, w_i \in \mathcal{T}_i$ we can get that $\mathbb{E}_{t_{-i}\sim\widehat{\mathcal{D}}_{-i}}\left[t_i\left(\mathcal{M}(\tau(t_i),t_{-i})\right)\right] \geq \mathbb{E}_{t_{-i}\sim\widehat{\mathcal{D}}_{-i}}\left[t_i\left(\widehat{\mathcal{M}}(t_i,t_{-i})\right)\right] - \beta n H$.

This concludes the proof of the intermediate bound. To conclude the proof for the BIC guarantee we need to show that:

$$\mathbb{E}_{t_{-i}\sim\widehat{\mathcal{D}}_{-i}}\left[u_i^{\mathcal{M}}(\tau(t_i) \leftarrow \tau(w_i),t_{-i})\right] \geq -4H\delta - \varepsilon.$$

By Proposition 5, $u_i^{\mathcal{M}}(\tau(t_i) \leftarrow \tau(w_i),t_{-i}) \in [-H,3H]$, for all $i \in [n], t_i, w_i \in \mathcal{T}_i, t_{-i} \in \mathcal{T}_{-i}$, and hence $u_i^{\mathcal{M}}(\tau(t_i) \leftarrow \tau(w_i),t_{-i}) - u_i^{\mathcal{M}}(\tau(t_i) \leftarrow \tau(w_i),t'_{-i}) \leq 4H\,\mathbb{1}\{t_{-i} \neq t'_{-i}\}$. Thus, for any coupling $\gamma$ of $\mathcal{D}_{-i}$ and $\widehat{\mathcal{D}}_{-i}$, and thus for the optimal coupling $\gamma^*$ between $\mathcal{D}_{-i}$ and $\widehat{\mathcal{D}}_{-i}$, we get

$$\mathbb{E}_{(t_{-i},t'_{-i})\sim\gamma^*}\left[u_i^{\mathcal{M}}(\tau(t_i) \leftarrow \tau(w_i),t_{-i}) - u_i^{\mathcal{M}}(\tau(t_i) \leftarrow \tau(w_i),t'_{-i})\right] \leq 4H\,d_{\mathsf{TV}}\left(\mathcal{D}_{-i},\widehat{\mathcal{D}}_{-i}\right)$$

$$\leq 4H\,d_{\mathsf{TV}}\left(\mathcal{D},\widehat{\mathcal{D}}\right)$$

$$\leq 3H\,\delta.$$

Using linearity of expectation and the fact that the chosen coupling maintains the marginals, by re-arranging we have:

$$-\mathbb{E}_{t'_{-i}\sim\widehat{\mathcal{D}}_{-i}}\left[u_i^{\mathcal{M}}(\tau(t_i) \leftarrow \tau(w_i),t'_{-i})\right] \leq 4H\,\delta - \mathbb{E}_{t_{-i}\sim\mathcal{D}_{-i}}\left[u_i^{\mathcal{M}}(\tau(t_i) \leftarrow \tau(w_i),t_{-i})\right]$$

$$\leq 4H\,\delta + \varepsilon,$$

where in the last inequality we used the fact that, since $\tau(t_i) \in \mathcal{T}_i^+$, from the definition of $\mathcal{M}$, for all $w_i, t_i \in \mathcal{T}_i$, we have $\mathbb{E}_{t_{-i}\sim\mathcal{D}_{-i}}\left[u_i^{\mathcal{M}}(\tau(t_i) \leftarrow \tau(w_i),t_{-i})\right] \geq -\varepsilon$.

We will now prove the revenue guarantee of the lemma. The tensorization property of TV distance [LPW09, Chapter 4] implies that $d_{\mathsf{TV}}\left(\widehat{\mathcal{D}},\tau(\widehat{\mathcal{D}})\right) \leq \beta n$, and thus from the triangle inequality, $d_{\mathsf{TV}}\left(\mathcal{D},\tau(\widehat{\mathcal{D}})\right) \leq \delta + \beta n$. Now notice from triangle inequality that $d_{\mathsf{TV}}\left(\mathcal{D},\tau(\widehat{\mathcal{D}})\right) \leq d_{\mathsf{TV}}\left(\mathcal{D},\widehat{\mathcal{D}}\right) + d_{\mathsf{TV}}\left(\widehat{\mathcal{D}},\tau(\widehat{\mathcal{D}})\right)$. Let $t \sim \mathcal{D}$ and $\widehat{t} \sim \tau(\widehat{\mathcal{D}})$. Since $d_{\mathsf{TV}}\left(\mathcal{D},\tau(\widehat{\mathcal{D}})\right) \leq \beta n + \delta$ there exists a coupling where $t \neq \widehat{t}$ with probability less than $\beta n + \delta$. Whenever $t = \widehat{t}$ the two mechanisms make exactly the same revenue. Whenever they are not, their difference is bounded by $V$. The desired inequality follows. □

Lemma 5 is then a simple corollary of Lemma 7.

*Proof of Lemma 5.* For an $(\varepsilon,q)$-BIC mechanism $\mathcal{M}$, one can split the type space $\mathcal{T}_i$ of each agent $i$ into two disjoint sets, $\mathcal{T}_i^G$ and $\mathcal{T}_i^B$, such that when $t_i \in \mathcal{T}_i^G$ agent $i$ $\varepsilon$-maximizes her utility by reporting $t_i$, and $\Pr_{t_i\sim\mathcal{D}}\left[t_i \in \mathcal{T}_i^B\right] \leq q$. Noting that $\perp \in \mathcal{T}_i^G$, the corollary is an immediate implication of Lemma 7. □

*Proof of Theorem 3.* The $(\varepsilon,q)$-BIC property is an immediate consequence of Lemma 4.

Applying Lemma 2, with $\mathcal{O}$ as the revenue objective (which is lower bounded by $-V/2$ and upper bounded by $V/2$), and setting $P = \mathcal{D}^p$, $Q = \mathcal{D}$, and $\mathcal{M} = \mathcal{M}_{\mathcal{D}^p}^a$, we have that $Rev(\mathcal{M}_{\mathcal{D}^p}^a,\mathcal{D}) \geq Rev(\mathcal{M}_{\mathcal{D}^p}^a,\mathcal{D}^p) - 2V\delta \geq \alpha\,OPT(\mathcal{D}^p) - 2V\delta$. Our main goal will be to lower bound $OPT(\mathcal{D}^p)$.

Let $\mathcal{M}_{\mathcal{D}}^*$ be the revenue optimal mechanism for $\mathcal{D}$. By Lemma 4, $\mathcal{M}_{\mathcal{D}}^*$ is an ex-post IR and $(\frac{8H\delta}{q}, q)$-BIC mechanism for $\mathcal{D}^p$ (for all $q \in [0,1]$). Therefore, Lemma 5 implies that there exists a mechanism $\widehat{\mathcal{M}}$ that is ex-post IR and $O(\frac{H\delta}{q} + nqH)$-BIC with respect to $\mathcal{D}^p$, such that $Rev(\widehat{\mathcal{M}}, \mathcal{D}^p) \geq Rev(\mathcal{M}_{\mathcal{D}}^*, \mathcal{D}^p) - nqV$.

Next, we apply the $\varepsilon$-BIC to BIC reduction of [COVZ21], on the mechanism $\mathcal{M}_{\mathcal{D}}^*$. Specifically, we use the following lemma.

**Lemma 8** ([DW12], [RW18], [COVZ21])**.** *In any $n$ agent setting where the valuations of agents are bounded by $H$, for any mechanism $\mathcal{M}$ with payments in $[-H, H]$, that is ex-post IR and $\varepsilon$-BIC with respect to some product distribution $\mathcal{D}$, there exists a mechanism $\mathcal{M}'$ with payments in $[-H, H]$, [1] that is ex-post IR and BIC with respect to $\mathcal{D}$, such that, assuming truthful bidding $Rev(\mathcal{M}', \mathcal{D}) \geq Rev(\mathcal{M}, \mathcal{D}) - O(n\sqrt{H\varepsilon})$.*

So, Lemma 8 implies that there exists a mechanism $\mathcal{M}'$ that is ex-post IR and BIC with respect to $\mathcal{D}^p$ such that $Rev(\mathcal{M}', \mathcal{D}^p) \geq Rev(\widehat{\mathcal{M}}, \mathcal{D}^p) - O(n\sqrt{H(\frac{H\delta}{q} + nqH)})$. Combining all the ingredients so far, we have

$$
\begin{aligned}
Rev(\mathcal{M}_{\mathcal{D}^p}^a, \mathcal{D}) &\geq Rev(\mathcal{M}_{\mathcal{D}^p}^a, \mathcal{D}^p) - V\delta \\
&\geq \alpha \, OPT(\mathcal{D}^p) - V\delta \\
&\geq \alpha \, Rev(\mathcal{M}', \mathcal{D}^p) - V\delta \\
&\geq \alpha \, Rev(\widehat{\mathcal{M}}, \mathcal{D}^p) - O\left(\alpha n\sqrt{H(\frac{H\delta}{q} + nqH)} + V\delta\right) \\
&\geq \alpha \, Rev(\mathcal{M}_{\mathcal{D}}^*, \mathcal{D}^p) - O\left(\alpha n\sqrt{H(\frac{H\delta}{q} + nqH)} + V(\delta + \alpha nq)\right) \\
&= \alpha \, Rev(\mathcal{M}_{\mathcal{D}}^*, \mathcal{D}^p) - O\left(\alpha nH\sqrt{\frac{\delta}{q} + nq} + V(\delta + \alpha nq)\right)
\end{aligned}
$$

Applying Lemma 2 again, with $P = \mathcal{D}$, $Q = \mathcal{D}^p$, and $\mathcal{M} = \mathcal{M}_{\mathcal{D}}^*$ we have $Rev(\mathcal{M}_{\mathcal{D}}^*, \mathcal{D}^p) \geq OPT(\mathcal{D}) - V\delta$. Combining with the previous inequality, we have $Rev(\mathcal{M}_{\mathcal{D}^p}^a, \mathcal{D}) \geq \alpha OPT(\mathcal{D}) - O\left(\alpha nH\sqrt{\frac{\delta}{q} + nq} + \alpha nqV + (1+\alpha)V\delta\right)$. Picking $q = \sqrt{\delta/n}$, and noting that $V \leq 2nH$, we have: $Rev(\mathcal{M}_{\mathcal{D}^p}^a, \mathcal{D}) \geq \alpha OPT(\mathcal{D}) - O\left(\alpha V(n\delta)^{1/4} + \alpha V(n\delta)^{1/2} + (1+\alpha)V\delta\right) \geq \alpha OPT(\mathcal{D}) - O\left((1+\alpha)V\sqrt{n\sqrt{\delta}}\right)$. □

*Proof of Proposition 1.* The marginal distributions for $\mathcal{D}^p$ and $\mathcal{D}$ are close in total variation distance, and specifically, $d_{\mathsf{TV}}\left(\widehat{\mathcal{D}}_i, \mathcal{D}_i^p\right) \leq d_{\mathsf{TV}}\left(\widehat{\mathcal{D}}, \mathcal{D}^p\right) \leq \varepsilon$. Therefore, $d_{\mathsf{TV}}\left(\mathcal{D}_i, \mathcal{D}_i^p\right) \leq \varepsilon$, which implies that $d_{\mathsf{TV}}\left(\mathcal{D}, \mathcal{D}^p\right) \leq n\varepsilon$. Applying the triangle inequality completes the proof. □

# C   Proofs missing from Section 4.2

*Proof of Theorem 4.* In order to prove this theorem we will first need to prove two intermediate lemmas. Recall that $\Pi(\mathcal{D}_1, \cdots, \mathcal{D}_n) = \{\mathcal{D}' | \Pr_{t_i \sim \mathcal{D}_i}[t_i = v_i] = \sum_{v_{-i} \in \mathcal{T}_{-i}} \Pr_{t \sim \mathcal{D}'}[t = (v_i, v_{-i})], \forall i \in [n], \forall t_i \in \mathcal{T}_i\}$.

**Lemma 9.** *For any distribution $\mathcal{D} \in \Pi(\mathcal{D}_1, \cdots, \mathcal{D}_n)$ there exists a distribution $\mathcal{D}' \in \Pi(\mathcal{D}_1', \cdots, \mathcal{D}_n')$ such that $d_{\mathsf{TV}}(\mathcal{D}, \mathcal{D}') \leq n\varepsilon$, where for all $i$, $d_{\mathsf{TV}}(\mathcal{D}_i, \mathcal{D}_i') \leq \varepsilon$.*

---

[1] In the reduction payments are only scaled by a value less than 1. Thus if $\mathcal{M}$ had payments in $[-H, H]$, then $\mathcal{M}'$ also has payments in that range.

*Proof.* We will prove an intermediate step that will then immediately yield the desired outcomes. More precisely we will first show that for any distribution $\mathcal{D}^{(i-1)} \in \Pi(\mathcal{D}'_1, \cdots, \mathcal{D}'_{i-1}, \mathcal{D}_i, \cdots \mathcal{D}_n)$ there exists a distribution $\mathcal{D}^{(i)} \in \Pi(\mathcal{D}'_1, \cdots, \mathcal{D}'_{i-1}, \mathcal{D}'_i, \cdots \mathcal{D}_n)$ such that $d_{\mathsf{TV}}\left(\mathcal{D}^{(i-1)}, \mathcal{D}^{(i)}\right) \leq \varepsilon$, where $d_{\mathsf{TV}}\left(\mathcal{D}_i, \mathcal{D}'_i\right) \leq \varepsilon$. To prove this we will leverage the $\mathcal{L}^1$-distance characterization of TV distance.

Our proof will be constructive through a simple "moving mass" argument. For simplicity let's assume that there exist $v_i, v'_i \in \mathcal{T}_i$ such that $\Pr_{t_i \sim \mathcal{D}_i}[t_i = v_i] = \Pr_{t'_i \sim \mathcal{D}'_i}[t'_i = v_i] + \varepsilon$ and $\Pr_{t_i \sim \mathcal{D}_i}[t_i = v'_i] = \Pr_{t'_i \sim \mathcal{D}'_i}[t'_i = v'_i] - \varepsilon$. Extending the following procedure for arbitrary $\mathcal{D}_i$, $\mathcal{D}'_i$ such that $d_{\mathsf{TV}}\left(\mathcal{D}_i, \mathcal{D}'_i\right) \leq \varepsilon$ will be immediate. Given $\mathcal{D}^{(i-1)}$, construct $\mathcal{D}^{(i)}$ as follows:

1. Set $\varepsilon_{cur} = \varepsilon$ and $\mathcal{D}^{(i-1)} = \mathcal{D}^{(i)}$.

2. As long as $\varepsilon_{cur} > 0$ do the following process:

    (a) Find $v_{-i} \in \mathcal{T}_{-i}$ such that $\Pr_{t' \sim \mathcal{D}^{(i)}}[t' = (v_i, v_{-i})] > 0$ and let $\gamma$ be the minimum of $\Pr_{t' \sim \mathcal{D}^{(i)}}[t' = (v_i, v_{-i})]$ and $\varepsilon_{cur}$.
    (b) Change $\mathcal{D}^{(i)}$ such that $\Pr_{t' \sim \mathcal{D}^{(i)}}[t' = (v_i, v_{-i})] - \gamma$ and $\Pr_{t' \sim \mathcal{D}^{(i)}}[t' = (v'_i, v_{-i})] + \gamma$.
    (c) Set $\varepsilon_{cur} = \varepsilon_{cur} - \gamma$

3. Output $\mathcal{D}^{(i)}$

From our construction of $\mathcal{D}^{(i)}$ it is immediate that $\mathcal{D}^{(i)} \in \Pi(\mathcal{D}'_1, \cdots, \mathcal{D}'_{i-1}, \mathcal{D}'_i, \cdots \mathcal{D}_n)$ and $d_{\mathsf{TV}}\left(\mathcal{D}^{(i-1)}, \mathcal{D}^{(i)}\right) \leq \varepsilon$. Chaining up the resulting inequalities and using triangle inequality concludes the proof. □

Leveraging the above we can prove the following:

**Lemma 10.** *For any mechanism $\mathcal{M}$ and sets of marginals $(\mathcal{D}_1, \cdots, \mathcal{D}_n)$ and $(\mathcal{D}'_1, \cdots, \mathcal{D}'_n)$ such that for all $i \in [n]$, $d_{\mathsf{TV}}\left(\mathcal{D}_i, \mathcal{D}'_i\right) \leq \varepsilon$ we have that:*

$$\min_{\mathcal{D} \in \Pi(\mathcal{D}_1, \cdots, \mathcal{D}_n)} \mathbb{E}_{t \sim \mathcal{D}}\left[\mathcal{O}(t, \mathcal{M}(t))\right] \geq \min_{\mathcal{D}' \in \Pi(\mathcal{D}'_1, \cdots, \mathcal{D}'_n)} \mathbb{E}_{t' \sim \mathcal{D}'}\left[\mathcal{O}(t', \mathcal{M}(t'))\right] - n\varepsilon V$$

*Proof.* We will prove this using a contradiction. Assume that

$$\min_{\mathcal{D} \in \Pi(\mathcal{D}_1, \cdots, \mathcal{D}_n)} \mathbb{E}_{t \sim \mathcal{D}}\left[\mathcal{O}(t, \mathcal{M}(t))\right] < \min_{\mathcal{D}' \in \Pi(\mathcal{D}'_1, \cdots, \mathcal{D}'_n)} \mathbb{E}_{t' \sim \mathcal{D}'}\left[\mathcal{O}(t', \mathcal{M}(t'))\right] - n\varepsilon V.$$

Lets call $\mathcal{D}^* = \arg\min_{\mathcal{D} \in \Pi(\mathcal{D}_1, \cdots, \mathcal{D}_n)} \mathbb{E}_{t \sim \mathcal{D}}\left[\mathcal{O}(t, \mathcal{M}(t))\right]$. Now using Lemma 9 we have that there exists $\widehat{\mathcal{D}}^* \in \Pi(\mathcal{D}'_1, \cdots, \mathcal{D}'_n)$ such that $d_{\mathsf{TV}}\left(\mathcal{D}^*, \widehat{\mathcal{D}}^*\right) \leq n\varepsilon$. Using Lemma 2 we have that $\mathbb{E}_{t \sim \mathcal{D}^*}\left[\mathcal{O}(t, \mathcal{M}(t))\right] \geq \mathbb{E}_{t \sim \widehat{\mathcal{D}}^*}\left[\mathcal{O}(t, \mathcal{M}(t))\right] - n\varepsilon V$. Chaining the above inequalities we get that:

$$\mathbb{E}_{t \sim \widehat{\mathcal{D}}^*}\left[\mathcal{O}(t, \mathcal{M}(t))\right] - n\varepsilon V \leq \mathbb{E}_{t \sim \mathcal{D}^*}\left[\mathcal{O}(t, \mathcal{M}(t))\right] < \min_{\mathcal{D}' \in \Pi(\mathcal{D}'_1, \cdots, \mathcal{D}'_n)} \mathbb{E}_{t' \sim \mathcal{D}'}\left[\mathcal{O}(t', \mathcal{M}(t'))\right] - n\varepsilon V$$

However, $\min_{\mathcal{D}' \in \Pi(\mathcal{D}'_1, \cdots, \mathcal{D}'_n)} \mathbb{E}_{t' \sim \mathcal{D}'}\left[\mathcal{O}(t', \mathcal{M}(t'))\right] - n\varepsilon V \leq \mathbb{E}_{t \sim \widehat{\mathcal{D}}^*}\left[\mathcal{O}(t, \mathcal{M}(t))\right] - n\varepsilon V$ which concludes the contradiction. □

Now we have all the components to prove the main theorem.

First by using Lemma 10 on $\mathcal{M}^\alpha$ we have that $\min_{\mathcal{D}' \in \Pi(\mathcal{D}'_1, \cdots, \mathcal{D}'_n)} \mathbb{E}_{t \sim \mathcal{D}'}\left[\mathcal{O}(t, \mathcal{M}^\alpha(t))\right] \geq \min_{\mathcal{D} \in \Pi(\mathcal{D}_1, \cdots, \mathcal{D}_n)} \mathbb{E}_{t \sim \mathcal{D}}\left[\mathcal{O}(t, \mathcal{M}^\alpha(t))\right] - n\varepsilon V$.

Now lets call $\mathcal{M}^* = \arg\max_{\mathcal{M}'} \min_{\mathcal{D}' \in \Pi(\mathcal{D}'_1, \cdots, \mathcal{D}'_n)} \mathbb{E}_{t \sim \mathcal{D}'}\left[\mathcal{O}(t, \mathcal{M}'(t))\right]$. By applying Lemma 10 on $\mathcal{M}^*$ we have that $\min_{\mathcal{D} \in \Pi(\mathcal{D}_1, \cdots, \mathcal{D}_n)} \mathbb{E}_{t \sim \mathcal{D}}\left[\mathcal{O}(t, \mathcal{M}^*(t))\right] \geq$

$\min_{\mathcal{D}'\in\Pi(\mathcal{D}_1',\cdots,\mathcal{D}_n')}\mathbb{E}_{t\sim\mathcal{D}'}\left[\mathcal{O}(t,\mathcal{M}^*(t))\right]$. Chaining all of the above we have that:

$$
\begin{aligned}
\min_{\mathcal{D}'\in\Pi(\mathcal{D}_1',\cdots,\mathcal{D}_n')}\mathbb{E}_{t\sim\mathcal{D}'}\left[\mathcal{O}(t,\mathcal{M}^\alpha(t))\right] &\geq \min_{\mathcal{D}\in\Pi(\mathcal{D}_1,\cdots,\mathcal{D}_n)}\mathbb{E}_{t\sim\mathcal{D}}\left[\mathcal{O}(t,\mathcal{M}^\alpha(t))\right] - n\varepsilon V \\
&\geq \alpha\max_{\mathcal{M}'}\min_{\mathcal{D}\in\Pi(\mathcal{D}_1,\cdots,\mathcal{D}_n)}\mathbb{E}_{t\sim\mathcal{D}}\left[\mathcal{O}(t,\mathcal{M}'(t))\right] - n\varepsilon V \\
&\geq \alpha\min_{\mathcal{D}\in\Pi(\mathcal{D}_1,\cdots,\mathcal{D}_n)}\mathbb{E}_{t\sim\mathcal{D}}\left[\mathcal{O}(t,\mathcal{M}^*(t))\right] - n\varepsilon V \\
&\geq \alpha\min_{\mathcal{D}'\in\Pi(\mathcal{D}_1',\cdots,\mathcal{D}_n')}\mathbb{E}_{t\sim\mathcal{D}'}\left[\mathcal{O}(t,\mathcal{M}^*(t))\right] - (1+\alpha)n\varepsilon V \\
&= \alpha\max_{\mathcal{M}'}\min_{\mathcal{D}'\in\Pi(\mathcal{D}_1',\cdots,\mathcal{D}_n')}\mathbb{E}_{t\sim\mathcal{D}'}\left[\mathcal{O}(t,\mathcal{M}'(t))\right] - (1+\alpha)n\varepsilon V.
\end{aligned}
$$

$\square$

# D   Proofs missing from Section 4.4

*Proof of Proposition 2.* Let $S_{\mathcal{D}}$ be the mechanism that implements the better of bundling and selling separately, as computed on a prior $\mathcal{D}$. $S_{\mathcal{D}^p}$ is a DISC and ex-post IR mechanism, and $Rev(S_{\mathcal{D}^p},\mathcal{D}^p) \geq \frac{1}{6}Rev(\mathcal{D}^p)$. Thus, applying Theorem 1 we have that $Rev(S_{\mathcal{D}^p},\mathcal{D}) \geq \frac{1}{6}\ Rev(\mathcal{D}) - \frac{7}{6}H\delta$. The mechanism $S_{\mathcal{D}^p}$ is either selling each item separately, or it is setting a posted price for the grand bundle. If the former case occurs, then running $S_{\mathcal{D}^p}$ on $\mathcal{D}$ makes (weakly) less revenue than $SRev(\mathcal{D})$; if the latter case occurs, running $S_{\mathcal{D}^p}$ on $\mathcal{D}$ makes (weakly) less revenue than $BRev(\mathcal{D})$. Therefore, we overall have that $Rev(S_{\mathcal{D}},\mathcal{D}) \geq Rev(S_{\mathcal{D}^p},\mathcal{D})$. Combining with the previous inequality we get $Rev(S_{\mathcal{D}},\mathcal{D}) \geq \frac{1}{6}\ Rev(\mathcal{D}) - \frac{7}{6}H\delta$. $\square$

**MRFs.**   We state some basic definitions for Markov Random Fields.

**Definition 5** (Markov Random Field [SK75],[KS80],[CO21])**.** *A Markov Random Field (MRF) is defined by a hypergraph $G = (V,E)$. Associated with every vertex $v \in V$ is a random variable $X_v$ taking values in some alphabet $\Sigma_v$, as well as a potential function $\psi_v : \Sigma_v \to \mathbb{R}$. Associated with every hyperedge $e \subseteq E$ is a potential function $\psi_e : \Sigma_e \to \mathbb{R}$. In terms of these potentials, we define a probability distribution $\mathcal{D}$ associating to each vector $\mathbf{c} \in \times_{v\in V}\Sigma_v$ probability $\mathcal{D}(\mathbf{c})$ satisfying: $\mathcal{D}(\mathbf{c}) \propto \prod_{v\in V} e^{\psi_v(c_v)} \prod_{e\in E} e^{\psi_e(\mathbf{c}_e)}$, where $\Sigma_e$ denotes $\times_{v\in e}\Sigma_v$ and $\mathbf{c}_e$ denotes $\{c_v\}_{v\in e}$.*

**Definition 6** ([CO21])**.** *Given a random variable/type $\mathbf{t}$ genarated by an MRF over a hypergraph $G = ([m],E)$, we define **weighted degree** of item $i$ as: $d_i := \max_{x\in\mathcal{T}}|\sum_{e\in E:i\in e}\psi_e(x_e)|$ and the **maximum weighted degree** as $\Delta := \max_{i\in[m]} d_i$.*

**Lemma 11** (Lemma 2[CO21])**.** *Let random variable $t$ be generated by an MRF. For any $i$ and any set $\mathcal{E} \subseteq \mathcal{T}_i$ and set $\mathcal{E}' \subseteq \mathcal{T}_{-i}$:*

$$
\exp(-4\Delta) \leq \frac{\Pr_{t\sim\mathcal{D}}\left[t_i \in \mathcal{E} \wedge t_{-i} \in \mathcal{E}'\right]}{\Pr_{t_i\sim\mathcal{D}_i}\left[t_i \in \mathcal{E}\right]\Pr_{t_{-i}\sim\mathcal{D}_{-i}}\left[t_{-i} \in \mathcal{E}'\right]}) \leq \exp(4\Delta)
$$

*Proof of Proposition 3.* Consider the case where $m = 2$. Assume that for each item there exist two possible valuations $A, B$. Consider the following distribution $\mathcal{D}$ of possible valuations. $\Pr_{(t_1,t_2)\sim\mathcal{D}}\left[(t_1,t_2) = (A,A)\right] = 1 - 2k + k^3$, $\Pr_{(t_1,t_2)\sim\mathcal{D}}\left[(t_1,t_2) = (A,B)\right] = \Pr_{(t_1,t_2)\sim\mathcal{D}}\left[(t_1,t_2) = (B,A)\right] = k - k^3$, $\Pr_{(t_1,t_2)\sim\mathcal{D}}\left[(t_1,t_2) = (B,B)\right] = k^3$. Notice that for any $0 < k < 1/2$ this is a valid distribution. Its TV distance from the product of its marginals is $2(k^2 - k^3) \leq 2k^2$. From Lemma 11 we have $\exp(-4\Delta) \leq \frac{\Pr_{(t_1,t_2)\sim\mathcal{D}}[t_1=B\wedge t_2=B]}{\Pr_{t_1\sim\mathcal{D}_1}[t_1=B]\cdot\Pr_{t_2\sim\mathcal{D}_2}[t_2=B]} = \frac{k^3}{k\cdot k} = k$, which implies that $\Delta \geq \frac{1}{4}log(\frac{1}{k})$. $\square$

We can prove the statement of Proposition 3 in a different way by constructing a distribution $\mathcal{D}$ that is close to a product distribution but the parameter $\Delta$ is arbitrarily large.

*Proof.* Let $\mathcal{D}^p$ be a product distribution such that $\mathcal{D}^p(t) = \frac{1}{Z}\prod_{v\in V} e^{\psi_v(t_v)}$ where $Z$ (known as the partition function) normalizes the values to ensure that $\mathcal{D}^p$ is a probability distribution. Consider the profile $t^*$ that happens with the smallest probability. Let that probability be $0 < \delta \leq \frac{1}{2}$. We have that

$$
\mathcal{D}^p(t^*) = \frac{1}{Z}\prod_{v\in V} e^{\psi_v(t_v^*)} = \delta \tag{4}
$$

We can construct a joint distribution $\mathcal{D}$ that is produced by an MRF in a way that the TV distance between $\mathcal{D}^p$ and $\mathcal{D}$ is bounded by $\delta$ while the parameter $\Delta$ of the MRF grows to infinity.

Let $\mathcal{D}(t) \propto \prod_{v \in V} e^{\widehat{\psi}_v(t_v)} \prod_{e \in E} e^{\psi_e(\mathbf{t}_e)}$ for some potential functions $\widehat{\psi}_v(\cdot)$ and $\psi_e(\cdot)$. We can construct $\mathcal{D}$ by selecting $\widehat{\psi}_v(t_v) = \psi_v(t_v)$ for all $v \in V$. Consider hyperedge $e^* = V$ (i.e. $e^*$ is the hyperedge that connects all nodes in $V$). For that hyperedge $e^*$ and the profile $t^*$ we choose $\psi_{e^*}(\mathbf{t}^*) \neq 0$, and for all other combinations of hyperedges $e$ and profiles $t_e$ we have that $\psi_e(\mathbf{t}_e) = 0$. We choose $\psi_{e^*}(\mathbf{t}^*)$ value such that $\mathcal{D}(t^*) = \epsilon$, for some $0 \leq \epsilon < \delta$. For ease of notation let $e^{\psi_{e^*}(\mathbf{t}^*)} = c(\epsilon)$. Let $Z'(\epsilon)$ be the partition function of $\mathcal{D}$, which depends on the choice of $\epsilon$. From the above, it is not difficult to see that $\forall t \neq t^* : \mathcal{D}(t) = \frac{1}{Z'(\epsilon)} \prod_{v \in V} e^{\psi_v(t_v)}$, and $\mathcal{D}(t^*) = \frac{1}{Z'(\epsilon)} \prod_{v \in V} e^{\psi_v(t_v)} e^{\psi_{e^*}(\mathbf{t}^*)} = \frac{1}{Z'(\epsilon)} \prod_{v \in V} e^{\psi_v(t_v)} \cdot c(\epsilon)$. Using Equation (4), we can rewrite $\mathcal{D}(t^*)$ as

$$\mathcal{D}(t^*) = \frac{1}{Z'(\epsilon)} \prod_{v \in V} e^{\psi_v(t_v^*)} e^{\psi_{e^*}(\mathbf{t}^*)} = \frac{Z}{Z'(\epsilon)} \cdot \delta \cdot c(\epsilon) = \epsilon. \tag{5}$$

By the definition of the partition function we have that $Z = \sum_{t \in \mathcal{T}} \prod_{v \in V} e^{\psi_v(t_v)}$, and $Z'(\epsilon) = \sum_{t \in T} \prod_{v \in V} e^{\psi_v(t_v)} \prod_{e \in E} e^{\psi_e(\mathbf{t}_e)} = \sum_{t \in \mathcal{T}: t \neq t^*} \prod_{v \in V} e^{\psi_v(t_v)} + \prod_{v \in V} e^{\psi_v(t_v^*)} \cdot c(\epsilon)$. Since $\mathcal{D}^p(t^*) = \delta$ the remaining probability for all profiles is $(1 - \delta)$, so for the first part of the sum we have $\sum_{t \in \mathcal{T}: t \neq t^*} \prod_{v \in V} e^{\psi_v(t_v)} = Z(1 - \delta)$. We can use again Equation (4) to simplify the second part of $Z'(\epsilon)$. Therefore, we have

$$Z'(\epsilon) = Z(1 - \delta) + Z \cdot \delta \cdot c(\epsilon) \tag{6}$$

Rearranging Equation (5) we have $Z \cdot \delta \cdot c(\epsilon) = \epsilon \cdot Z'(\epsilon)$. Substituting that into Equation (6) we get that $Z'(\epsilon) = Z \frac{1-\delta}{1-\epsilon}$. Using the last formula back into Equation (5) we get that $c(\epsilon) = \frac{(1-\delta)\epsilon}{(1-\epsilon)\delta}$. As we take the probability $\mathcal{D}(t^*)$ to zero we have $\lim_{\epsilon \to 0} c(\epsilon) = \frac{(1-\delta)\epsilon}{(1-\epsilon)\delta} = 0$, and $\lim_{\epsilon \to 0} Z'(\epsilon) = \frac{Z(1-\delta)}{1-\epsilon} = Z(1 - \delta)$. Therefore, the distribution $\mathcal{D}$ behaves nicely as we take the probability of $t^*$ to zero. By Definition 6, $\Delta(\epsilon) = |\psi_{e^*}(\mathbf{t}^*)|$ since it is the only non-zero value of the potential function $\psi_e(\cdot)$. By definition $e^{\psi_{e^*}(\mathbf{t}^*)} = c(\epsilon) \implies \psi_{e^*}(\mathbf{t}^*) = \ln(c(\epsilon))$. Taking again $\epsilon$ to zero we can show that $\Delta(\epsilon)$ goes to infinity, $\lim_{\epsilon \to 0} \Delta(\epsilon) = \lim_{\epsilon \to 0} \ln(c(\epsilon)) = -\infty$.

We can calculate the TV distance:

$$\begin{aligned}
2\,d_{\mathsf{TV}}\left(\mathcal{D}, \mathcal{D}^p\right) &= \sum_{t \in T} |\mathcal{D}(t) - \mathcal{D}^p(t)| \\
&= \sum_{t \in T: t \neq t^*} |\mathcal{D}(t) - \mathcal{D}^p(t)| + |\mathcal{D}(t^*) - \mathcal{D}^p(t^*)| \\
&= \sum_{t \in T: t \neq t^*} \left| \frac{1}{Z} \prod_{v \in V} e^{\psi_v(t_v)} - \frac{1}{Z'(\epsilon)} \prod_{v \in V} e^{\psi_v(t_v)} \right| + \delta - \epsilon \\
&= \left| 1 - \frac{Z}{Z'(\epsilon)} \right| \sum_{t \in T: t \neq t^*} \left| \frac{1}{Z} \prod_{v \in V} e^{\psi_v(t_v)} \right| + \delta - \epsilon \\
&= \left| 1 - \frac{1-\epsilon}{1-\delta} \right| (1 - \delta) + \delta - \epsilon \\
&= 2(\delta - \epsilon)
\end{aligned}$$

To go from line 5 to line 6 we use the fact that $Z'(\epsilon) = Z \frac{1-\delta}{1-\epsilon}$ and that the sum of the probabilities acording to $\mathcal{D}^p$ of all the profiles except $t^*$ is $1 - \delta$.

That concludes the proof that there exists a distribution $\mathcal{D}$ that is at most $\delta$ away in TV from a product distribution for which the parameter $\Delta$ is unbounded. $\qquad\square$

*Proof of Proposition 4.* As a first step, we are going to bound the Kullback-Leibler (KL) divergence between the distribution $\mathcal{D}$ and a product distribution $\mathcal{D}^p$. Then we are going to use Pinsker's inequality [Tsy08] and the Bretagnolle-Huber inequality [Tsy08, BH78] to bound the TV distance using KL divergence.

Let $\mathcal{D}(t) = \frac{1}{Z_1} \prod_{v \in V} e^{\psi_v(t_v)} \prod_{e \in E} e^{\psi_e(t_e)}$, where $Z_1$ is the partition function. Let $\mathcal{D}^p$ be product distribution such that $\mathcal{D}^p(t) = \frac{1}{Z_2} \prod_{v \in V} e^{\psi_v(t_v)}$, where $Z_2$ is the partition function.

The KL divergence is between $\mathcal{D}$ and $\mathcal{D}^p$ is:

$$
\begin{aligned}
D_{KL}(\mathcal{D} || \mathcal{D}^p) &= \sum_{t \in \mathcal{T}} \mathcal{D}(t) \log \frac{\mathcal{D}(t)}{\mathcal{D}^p(t)} \\
&= \sum_{t \in \mathcal{T}} \mathcal{D}(t) \log \frac{Z_2 \prod_{v \in V} e^{\psi_v(t_v)} \prod_{e \in E} e^{\psi_e(t_e)}}{Z_1 \prod_{v \in V} e^{\psi_v(t_v)}} \\
&= \sum_{t \in \mathcal{T}} \mathcal{D}(t) \log \frac{Z_2}{Z_1} \prod_{e \in E} e^{\psi_e(t_e)} \\
&= \sum_{t \in \mathcal{T}} \mathcal{D}(t) \left( \log \frac{Z_2}{Z_1} + \sum_{e \in E} \psi_e(t_e) \right) \\
&\leq \sum_{t \in \mathcal{T}} \mathcal{D}(t) \left( \log \frac{Z_2}{Z_1} + \frac{m}{2} \Delta \right) \\
&= \frac{m}{2} \Delta + \log \frac{Z_2}{Z_1}
\end{aligned}
$$

Since KL divergence is not symmetric, we can also compute: $D_{KL}(\mathcal{D}^p || \mathcal{D})$:

$$
\begin{aligned}
D_{KL}(\mathcal{D}^p || \mathcal{D}) &= \sum_{t \in \mathcal{T}} \mathcal{D}^p(t) \log \frac{\mathcal{D}^p(t)}{\mathcal{D}(t)} \\
&= \sum_{t \in \mathcal{T}} \mathcal{D}(t) \log \frac{Z_1 \prod_{v \in V} e^{\psi_v(t_v)}}{Z_2 \prod_{v \in V} e^{\psi_v(t_v)} \prod_{e \in E} e^{\psi_e(t_e)}} \\
&= \sum_{t \in \mathcal{T}} \mathcal{D}(t) \log \frac{Z_1}{Z_2} \prod_{e \in E} e^{-\psi_e(t_e)} \\
&= \sum_{t \in \mathcal{T}} \mathcal{D}(t) \left( \log \frac{Z_1}{Z_2} - \sum_{e \in E} \psi_e(t_e) \right) \\
&\leq \sum_{t \in \mathcal{T}} \mathcal{D}(t) \left( \log \frac{Z_1}{Z_2} + \frac{m}{2} \Delta \right) \\
&= \frac{m}{2} \Delta - \log \frac{Z_2}{Z_1}
\end{aligned}
$$

We can get that $\sum_{e \in E} \psi_e(t_e) \in \left[ -\frac{m}{2} \Delta, \frac{m}{2} \Delta \right]$ as follows. $\sum_e \psi_e(t_e) = \frac{1}{2} \sum_{i \in [m]} \sum_{e \in E : i \in e} \psi_e(t_e) \leq \frac{1}{2} \sum_{i \in [m]} d_i \leq \frac{m \Delta}{2}$. Similarly, we can lower bound $\sum_{e \in E} \psi_e(t_e) \geq -\frac{m \Delta}{2}$ since the definition of $d_i$ is $d_i := \max_{x \in \mathcal{T}} | \sum_{e \in E : i \in e} \psi_e(x_e) |$.

From the above inequalities we have that $\min \{ D_{KL}(\mathcal{D}^p || \mathcal{D}), D_{KL}(\mathcal{D} || \mathcal{D}^p) \} \leq \frac{m}{2} \Delta$. From Pinsker's inequality we get $d_{\mathsf{TV}}(\mathcal{D}, \mathcal{D}^p) \leq \sqrt{\frac{m \Delta}{4}}$, and from the Bretagnolle-Huber inequality we get $d_{\mathsf{TV}}(\mathcal{D}, \mathcal{D}^p) \leq \sqrt{1 - \exp(-m \Delta / 2)}$. Combining these inequalities we have the desired bound on the TV distance. □