# OpenReview forum: "On the Robustness of Mechanism Design under Total Variation Distance"
_NeurIPS.cc/2023/Conference — NeurIPS 2023 poster_

### Official Review · Reviewer_JheU · 2023-07-04

**Soundness:** 4 excellent
**Presentation:** 3 good
**Contribution:** 3 good
**Rating:** 6
**Confidence:** 3

**Summary:**

The paper studies the following question: in what way can approximation guarantees of (approximately) IC mechanisms be preserved when the prior distribution is perturbed by a small amount in the total variation distance?  The main technical lemmas state that the guarantees of DSIC and BIC mechanisms are in fact preserved in ways that can be useful.  Based on these lemmas, the authors reproduce a number of previously known results, and also prove some new ones, in various subareas of algorithmic mechanism design.

**Strengths:**

The general question the paper aims to answer is very natural and important.  The main technical lemmas turn out to be quite powerful, although they are fairly simple and intuitive (which I think is good).  I like the way the various implications are derived in the paper, i.e., the right technical observations (in the case of this paper, the main technical lemmas in Section 3) can make things much easier.

**Weaknesses:**

One might complain that the main technical lemmas are not all that surprising, but I think the way the authors make use of these lemmas outweighs such criticism.  Another minor thing is the applications are relatively loosely organized, and I'd be more excited if there is a key application that irrefutably shows the power of the framework proposed in the paper.

**Questions:**

(also including detailed comments here)

Line 263, "two conditional P_{X, Y}, Q_{X, Y}": do you mean "joint"?

Lemma 3: could mention this is essentially a Markov-like bound (right?)

Lemma 5: some intuition might be helpful

Line 316: "Bei et. al." => "Bei et al." (consider using \citet or \citeauthor?)

Corollary 1: why do you need the mechanism to be posted pricing?

---

> ### Author Rebuttal · Authors · 2023-08-04
>
> Thank you for your thorough review and questions. Below we answer your questions.
>
> - "Line 263, "two conditional P_{X, Y}, Q_{X, Y}": do you mean "joint"?"
> - "Line 316: "Bei et. al." => "Bei et al." (consider using \citet or \citeauthor?)"
>
> We will fix both typos.
>
> - Lemma 3: could mention this is essentially a Markov-like bound (right?)
>
> We will mention above the lemma that it is a Markov-like result. To clarify, the lemma requires two steps to prove (as shown in the Supplementary Materials): 1) An argument to bound expected TV distance between conditional distributions using the TV distance between joint distributions, which is the main calculation, and 2) Markov’s inequality (as you correctly point out).
>
>
> - "Lemma 5: some intuition might be helpful"
>
> Intuitively, one can focus on the subset of “good” types such that reporting truthfully on M is within $\epsilon$ of the optimal report; a type will be “good” with probability $1-q$, which implies a bound on the TV distance between the original distribution and the distribution “restricted” to “good” types. Our construction alters the behavior of $M$ on the “bad” types, and uses the aforementioned TV bound and Lemma 2 to bound the additional loss in the BIC constraint, as well as the revenue loss. We’d be happy to include this intuition, as well as further expand/clarify if the reviewer believes it would improve the quality of the paper.
>
> - "Corollary 1: why do you need the mechanism to be posted pricing?"
>
> The corollary holds for all DSIC and ex-post IR mechanisms. We refer to posted prices here because they are the main family of policies studied in the literature on prophet inequalities. Specifically, the optimal policy and, to the best of our knowledge, every simple and approximately optimal policy in the literature are posted prices.

---

> > ### Comment · Reviewer_JheU · 2023-08-18
> >
> > Thank you for your response!  I have no further questions.

---

### Official Review · Reviewer_K563 · 2023-07-04

**Soundness:** 3 good
**Presentation:** 3 good
**Contribution:** 3 good
**Rating:** 7
**Confidence:** 3

**Summary:**

This paper studies the robust mechanism design. In this problem, there is a set of items and a set of agents whose valuation functions are drawn from a batch of unknown distributions. These distributions are correlated, i.e., they are close to a known distribution under the total variance distance. The agents' valuation functions are private information, and the goal is to design a truthful mechanism that maximizes some objective functions in expectation. The two main objectives are considered in this paper: social welfare maximization and revenue maximization.

The main contribution of this paper is: they prove dominant strategy incentive-compatible mechanisms are robust, namely alpha-approximate mechanisms under the classical setting can be converted into a mechanism under the robust setting without losing any factor on approximation ratio. A similar result holds for Beyseian incentive-compatible mechanisms, but a small factor has to be loss on the approximation ratio. Finally, the authors also list a batch of applications of the proposed framework.

**Strengths:**

1. I appreciated that the submission is carefully written and structured, so reads well given the technicality of the material. Especially, the flow of the paper is well-designed. The presentation of the algorithmic idea is also clear.

2. The studied problem is interesting and well-motivated.

3. The proposed framework works for a large number of applications, although the fundamental technical results are simple.

**Weaknesses:**

From my perspective, there are no major weaknesses, but there seems to be no big surprise in the used techniques. However, I am not in a good position on judging the technique novelty of this paper.

**Questions:**

I don't have any specific questions.

**Limitations:**

This is a theoretical paper, there is no potential negative societal impact.

---

> ### Author Rebuttal · Authors · 2023-08-03
>
> Thank you for your thorough review.

---

### Official Review · Reviewer_yCAy · 2023-07-05

**Soundness:** 3 good
**Presentation:** 3 good
**Contribution:** 3 good
**Rating:** 7
**Confidence:** 3

**Summary:**

This paper studies the robust design of mechanisms for a designer with general bounded objective, when the true distribution of the agent types (possibly correlated) is not the actual distribution. More precisely, the main idea is that the optimal incentive compatible mechanism designed for the a priori distribution approximates well the optimal mechanism for the true distribution as a function of the TV distance between the two distributions, as well as guarantees approximate incentive compatibility. In this way, it generalizes results from the existing literature which were focused on specific objectives such as welfare or revenue, and under product distribution. This work is decomposed in 2 main different parts, first results relating how the various metrics (objective and incentive compatibility) degrade in the TV distance for both DSIC and BIC mechanisms are presented, then these approximation results are used for applications such as approximations in the prophet inequality setting when the distributions may be correlated, or for approximation results on simple mechanisms.

**Strengths:**

- This paper study the important setting of mechanisms robust to small perturbations of the agents types distribution. It generalizes some previous results, and presents a variety of tools that can be useful for a mechanism designer. Multiple applications are given. Moreover the various applications, beyond their own interest, also serve as an example on how to apply these tools.
- The paper is clearly written, and the existing literature well presented. The link between previous results and how they are being generalized is transparent.
- I have went through the proofs in the main paper, as well as some in the appendix, and found no issues.

**Weaknesses:**

- Compared to other works, such as `Posted Pricing and Prophet Inequalities with Inaccurate Priors' (Dutting et al 2019), this paper only studies the TV distance.
- The approximation results are not related to any upper bound, which makes it difficult to evaluate the tightness of these results.

**Questions:**

- l638 : Does 'single agent' described in this context mean that $n=1$? In this case what would the product distribution $\mathcal{D}^p$ signify?
- The proof of Lemma $2$ uses a coupling argument to bound the difference between objectives under different distributions in terms of TV distance. Can similar coupling arguments be used to derive similar robustness results, but this time for Wasserstein distance? More generally, does it look possible to extend those results to more general distances (or f-divergences like the Kullback-Leibler) or are these results stemming from the specific properties of the TV distance?
- Is there an example when some of the proposed approximation bounds are tight, for instance in Theorem $1$?

**Limitations:**

The authors have correctly addressed some of the limitations of their work, such as discussing when some assumptions may be less general than previous works (common support of distributions necessary for Theorem 2, and weaker BIC guarantees).

---

> ### Author Rebuttal · Authors · 2023-08-04
>
> Thank you for the thoughtful review and questions. Below we provide answers to your questions.
>
> - "l638 : Does 'single agent' described in this context mean that n=1? In this case what would the product distribution signify?"
>
> Indeed this is the case of a single agent. The product distribution here is referring to the items. That is, the values for the items are drawn from independent distributions. We will clarify this.
>
> - "The proof of Lemma 2 uses.. the TV distance?"
>
> For the KL divergence (as well as many other f-divergences, such as $\chi^2$-divergence), we can readily use our results since a bound on KL implies a bound on TV (e.g., using Pinsker's inequality). Note that f-divergences also exhibit supremum characterizations like TV distance (using convex conjugation ideas), but these characterizations do not have the simple structure that TV exhibits. So, it's difficult to obtain crisp robustness results from these characterizations for general f-divergences. The easier approach is to bound TV using different f-divergences and use our results for TV distance as mentioned above.
>
> For Prokhorov distance, consider the following situation, noting that a bound on Prokhorov implies a bound on Wasserstein distance. Distribution $D$ is a point mass at $H$ (i.e. $H$ happens with probability 1). Distribution $D'$ is a point mass at $H-\delta$. The two distributions have a Prokhorov distance of $\delta$ but TV distance of 1. Now consider the single-item mechanism $M$ that posts a price of $H$. If the bidder that participates in $M$ draws her value for the item from $D$ she will always buy the item; thus the mechanism has expected revenue $H$. If the bidder draws her value from $D'$, $M$ has expected revenue equal to 0.
>
> This example demonstrates that our results cannot be immediately transferred to other Prokhorov or Wasserstein-type distances (despite the existence of coupling and/or duality characterizations), and “robustifying” a mechanism, as Brustle et al. do, seems necessary.
>
> - "Is there an example when some of the proposed approximation bounds are tight, for instance in Theorem 1?"
>
> Yes. See the “global” response.

---

> ### Comment · Reviewer_yCAy · 2023-08-13
>
> We thank the authors for their response.
>
> If possible, I would be happy to see if this tight example mentioned in the global response can be generalized beyond revenue or welfare, as the approximation bounds presented in this paper hold for more general objectives. Similarly, even if it is difficult to get tight examples for Theorem 2, I think it would be nice to have some partial negative results to show that these bounds are still not too bad.
>
> Otherwise, all my questions have been correctly addressed, and I believe this paper should be accepted as the contributions are novel and the writing and motivations are clear.

---

> > ### Author Response · Authors · 2023-08-16
> >
> > Thank you for your reply.
> >
> > The main issue with generalizing to arbitrary objectives is that a worst-case arbitrary objective can do something uninteresting, e.g. take the value c no matter what, where obviously our result is not tight. It is known that $E_P[f(X)] - E_Q[f(X)] \leq TV(P,Q)$ for all functions f bounded by 1/2, and equality holds for the function $f^*(x) = 1/2$ if $P(x) \geq Q(x)$ and $f^*(x) = -1/2$ otherwise. We can use this to show that equality will hold for Lemma 2 when the objective function and the mechanism, combined, look like this function, i.e., $f^*(x) = \mathcal{O}(x,M(x))$, with appropriate re-scaling when V is not 1. This would give a non-trivial sufficient condition for functions beyond welfare and revenue.
> >
> > We will also include partial bounds for Theorem 2.

---

### Official Review · Reviewer_XhLk · 2023-07-06

**Soundness:** 3 good
**Presentation:** 3 good
**Contribution:** 3 good
**Rating:** 6
**Confidence:** 4

**Summary:**

The paper studies a robust auction design problem for multiple items. In the model, the authors assume that they can access a (predicted) valuation distribution over all agents, and the total variance between the predicted distribution and the actual distribution is bounded. The goal is to design a truthful mechanism such that given any objective, the expected performance is robust with respect to the total variance bound.

Both DSIC and BIC mechanisms are considered in the paper. For the DISC setting, the paper shows that when the total variance is at most $\delta$ and the length of the range of the objective function is at most $V$, if there exists a $\alpha$-approximate mechanism (for the special case that $\delta=0$), using the mechanism directly can return an expected objective at least $\alpha OPT- (1+\alpha) V \delta$. For the BIC setting, the authors prove a similar theorem. They show that the difference between the objective obtained by a mechanism on two close valuation distributions is at most $V\delta$. Several applications are mentioned in the paper. The authors illustrate how their ideas can be applied to these concrete applications.

**Strengths:**

The paper extends the previous result on robust auction design and shows that for any objective function, once the range is bounded, we can obtain a robust mechanism with respect to the total variance easily. This result is interesting, and the basic idea might be useful in many other mechanism design settings.

**Weaknesses:**

One shortcoming of the proposed result is that the mechanism still does not have a performance guaranteed when the total variance is large. Maybe the authors could borrow some ideas from the learning-augmented algorithms and find an efficient way to combine the predicted mechanism and the traditional worst-case mechanism, such that the mechanism is still competitive even if the total variance is large.



**Questions:**

Is there any negative result for the model? For example, could you give a concrete hard instance such that the difference is at least $V\delta$?

**Limitations:**

See weakness.

---

> ### Author Rebuttal · Authors · 2023-08-03
>
>
> Thank you for the thoughtful review and question. Indeed, when the total variation is large, our results don’t have any bite, which, of course, does not imply that some sort of robustness is impossible to show. The approach of using learning-augmented algorithms that the reviewer suggests is a great direction for future work.
>
> Regarding your question, “Is there any negative result for the model? For example, could you give a concrete hard instance such that the difference is at least $V \delta$?”, see the “global” response.

---

> > ### Comment · Reviewer_XhLk · 2023-08-16
> >
> > I have gone through the hard instance stated in the global rebuttal. My question has been addressed.

---

### Official Review · Reviewer_acoM · 2023-07-11

**Soundness:** 3 good
**Presentation:** 2 fair
**Contribution:** 3 good
**Rating:** 5
**Confidence:** 3

**Summary:**

This paper considers the design and performance guarantees of various mechanisms under prior distributions, and aims to provide a general account of what happens to these mechanisms and their guarantees when these (joint) prior distributions are perturbed. They use the definition of TV distance in terms of the largest difference in probabilities over all events and combine it with the assumption that the mechanisms in question have bounded values and payments in order to argue that expected values, rewards, and incentives are only incrementally affected by small perturbations in TV distance. This observation allows for the 'robustificaton' of a number of prior results. The authors present some more technically involved claims for settings involving product prior distributions and Markov random fields.

**Strengths:**

This work considers a natural question, and aims to provide a systematic answer.

The applications and results are a combination of recovering prior robustness results and extending prior non-robust mechanisms and results to nearby joint prior distributions.

**Weaknesses:**

Upon some reflection it makes sense that the robustness guarantees the authors consider should contribute additive terms which depend on the maximum payment or reward in a mechanism. But for the bounds presented there are frequently other factors in the additive terms. This paper would benefit from discussion of lower bounds, and more generally from arguments that the forms of these guarantees make sense.

Many of the claims are not stated along with all of the relevant conditions and caveats, for instance restrictions on boundedness and on the supports of distributions.

The proofs of some central claims are intuitive and straightforward, and at the same time many claims are lacking outlines of their proofs. The authors herald their usage of a "fundamental duality property of total variation distance," but it is unclear in which proofs this duality is being used, or the sense in which it is being employed.

**Questions:**

Where is the crux of your invocation of duality, and how are you using the connection between these characterizations of TV distance?

Which of your robustness claims readily admit lower (upper?) bounds?


Possible corrections:
L195-196: Should Omega additionally be a measurable subset in order to be a standard Borel space?
L213: is the supremum necessary?
L244: "inequality is because"
L310: epsilon used before it is introduced
L376: "distributions"
L384: is the Omega here intended to mean that there is some constant for which the claim holds?

**Limitations:**

Some of the theorem statements seem to be missing quantifiers and conditions, as well as informal overviews of their methods of proof.

It would be helpful to have more discussion of which dependences are necessary.

---

> ### Author Rebuttal · Authors · 2023-08-04
>
> Thank you for the thoughtful review and questions. We will update our theorems and lemmas, whenever applicable, to state the relevant conditions/restrictions necessary.
>
> Below we answer your questions.
>
> - Where is the crux of your invocation of duality, and how are you using the connection between these characterizations of TV distance?
>
> The key duality property of TV distance refers to the fact that it has two equivalent characterizations: 1) the maximum over bounded functions of the difference of expectations, and 2) the minimum over couplings of the probability that the two associated random variables are different. This property is presented in Lemma 1, and it is distilled in Lemma 2 in the context of mechanism design. Specifically, the result in Lemma 2 is really a form of (weak) duality that tells us that the difference between expected objective functions is bounded by TV distance. Its proof illustrates how we use the minimum coupling characterization of TV distance from Lemma 1.
>
> This means that we are invoking duality of TV distance anywhere we are using Lemma 2. Lemma 2 is crucially used in both Theorem 1 (DSIC robustness) and in Theorem 2 (BIC robustness). Moreover, in the applications, we use it in Corollary 1 (Prophet inequality) and marginal robustness (Lemma 10, towards proving Theorem 4), in Lemma 7 in order to prove Lemma 5 ($(\epsilon,q)$-BIC to $(\epsilon + nqH)$-BIC reduction), and in the proof of Theorem 3 (application for BIC mechanisms).
>
> Note also that although the duality property of TV distance is related to notions like linear programming duality, we do not use any such form of duality to prove Lemma 2. Rather, Lemma 2 itself is proving a form of duality. We hope this clarifies our use of “duality”, and we are also happy to clarify this in the paper.
>
> - Which of your robustness claims readily admit lower (upper?) bounds?
>
> See the “global” response for an answer to this question.
>
> - L195-196: Should Omega additionally be a measurable subset in order to be a standard Borel space?
>
> Yes, $\Omega$ must be measurable. This will be clarified in the paper.
>
> - L213: is the supremum necessary?
>
> The supremum used in the dual characterization of TV distance could be replaced by a maximum, i.e., the maximizing function exists under our assumptions. This will be changed in the paper. (The supremum used to define the bound of 1/2 on the functions is necessary.)
>
> - L244: "inequality is because". L310: epsilon used before it is introduced. L376: "distributions".
>
> Thank you for catching these; we fixed them.
>
> - L384: is the Omega here intended to mean that there is some constant for which the claim holds?
>
> Correct.

---

> > ### Comment · Area_Chair_4ke6 · 2023-08-18
> >
> > Dear authors,
> >
> > Your message has been noted. The decision on your paper will be based on my discussion with the reviewers.
> > We will reach out to your should we require further clarifications.
> >
> > Regards,

---

### Author Rebuttal · Authors · 2023-08-04

We thank all reviewers for the thorough reviews and helpful comments. We will incorporate the valuable suggestions from all reviewers in the final version of this paper.

A common question that reviewers acoM, XhLk, and yCAy ask is whether our various bounds are tight. One can construct trivial tight examples for Lemma 2 and Theorem 1, at least for the case of revenue and welfare. For instance, for the case of a single agent, letting $\mathcal{T} = [0,V]$, consider the case that the distribution P is a point mass at V, and distribution Q takes the value V with probability $1-\delta$, and zero otherwise. The TV distance between P and Q is exactly $\delta$. Now, consider the simple mechanism M that posts a price of V. Its revenue/welfare under P is V, and its revenue under Q is $(1-\delta)V$. One can construct such simple examples to show tightness for Lemma 2 and Theorem 1 in general. For BIC (e.g. Thm 2) it seems slightly trickier to construct tight examples (where one loses both in the BIC constraint and the revenue).

We’d be happy to include such lower-bound examples in the final version of the paper.

---

### Decision · Program_Chairs · 2023-09-21

**Decision:**

Accept (poster)

**Comment:**

The reviewers agreed that this paper studies a natural question and found it interesting.